# PYKT: A Python Library to Benchmark Deep Learning based Knowledge Tracing Models

**Zitao Liu**
Guangdong Institute of Smart Education
Jinan University
Guangzhou, Guangdong, China
liuzitao@jnu.edu.cn

**Qiongqiong Liu**
Think Academy International Education
TAL Education Group
Beijing, China
liuqiongqiong1@tal.com

**Jiahao Chen**
Think Academy International Education
TAL Education Group
Beijing, China
chenjiahao@tal.com

**Shuyan Huang**[*]
Think Academy International Education
TAL Education Group
Beijing, China
huangshuyan@tal.com

**Jiliang Tang**
Computer Science and Engineering Department
Michigan State University
East Lansing, MI, United States
tangjili@msu.edu

**Weiqi Luo**
Guangdong Institute of Smart Education
Jinan University
Guangzhou, Guangdong, China
lwq@jnu.edu.cn

## Abstract

Knowledge tracing (KT) is the task of using students' historical learning interaction data to model their knowledge mastery over time so as to make predictions on their future interaction performance. Recently, remarkable progress has been made of using various deep learning techniques to solve the KT problem. However, the success behind deep learning based knowledge tracing (DLKT) approaches is still left somewhat unknown and proper measurement and analysis of these DLKT approaches remain a challenge. First, data preprocessing procedures in existing works are often private and custom, which limits experimental standardization. Furthermore, existing DLKT studies often differ in terms of the evaluation protocol and are far away real-world educational contexts. To address these problems, we introduce a comprehensive python based benchmark platform, PYKT, to guarantee valid comparisons across DLKT methods via thorough evaluations. The PYKT library consists of a standardized set of integrated data preprocessing procedures on 7 popular datasets across different domains, and 10 frequently compared DLKT model implementations for transparent experiments. Results from our fine-grained and rigorous empirical KT studies yield a set of observations and suggestions for effective DLKT, e.g., wrong evaluation setting may cause label leakage that generally leads to performance inflation; and the improvement of many DLKT approaches is minimal compared to the very first DLKT model proposed by Piech et al. [25]. We have open sourced PYKT and our experimental results at https://pykt.org/. We welcome contributions from other research groups and practitioners.

---

[*]The corresponding author: Shuyan Huang.

36th Conference on Neural Information Processing Systems (NeurIPS 2022) Track on Datasets and Benchmarks.

# 1 Introduction

The increasingly digitalized education tools and the popularity of online learning have produced an unprecedented amount of data that provides us with invaluable opportunities for applying AI in education. Knowledge tracing (KT) is the task of *using students' historical learning interaction data to model their knowledge mastery over time so as to make predictions on their future interaction performance*. Such predictive capabilities can potentially help students learn better and faster when paired with high-quality learning materials and instructions, which is crucial for building next-generation smart and personalized education.

The KT related research has been studied since 1990s where Corbett and Anderson, to the best of our knowledge, were the first to estimate students' current knowledge with regard to each individual knowledge component (KC) [8]. A KC is a description of a mental structure or process that a learner uses, alone or in combination with other KCs, to accomplish steps in a task or a problem[2]. Since then, many attempts have been made to solve the KT problem, such as probabilistic graphical models [13] and factor analysis based models [3, 14, 35]. Recently, due to the rapid advances of deep neural networks, deep learning based knowledge tracing (DLKT) models have become the de facto KT framework for modeling students' mastery of KCs [1, 7, 10, 12, 15, 16, 19, 21, 22, 23, 24, 25, 29, 30, 31, 36, 37, 44, 45, 46, 47].

Although DLKT approaches have constituted new paradigms of the KT problem [10, 22, 25, 30, 46] and achieved promising results, recent studies [15, 16, 25, 34] seem to resemble each other with very limited nuances from the methodological perspective. Most existing work only provides coarse evaluation and both the contributing factors leading to the success of DLKT and how the DLKT models perform in the real-world educational contexts still remain somewhat unknown. Furthermore, evaluations of existing DLKT work are not standardized and reported AUC results of the same approach on the same dataset vary surprisingly from 0.709 to 0.86 [2] (details discussed in Section 3.3). Therefore, there is a substantial need for a standardized DLKT benchmark platform, which ensures that methods can be compared in a fair and transparent manner. Researchers need to be able to evaluate their proposed approaches against a wide range of state-of-the-art (SOTA) methods on both publicly available and private datasets and practitioners need to be capable of differentiating advantages and disadvantages of the DLKT algorithms in real-world educational contexts.

In order to accelerate research in building advanced KT approaches, we systematically design PYKT, a fine-grained python based benchmark library that brings us closer to the requirements of real-world KT applications in educational contexts. Overall this paper makes the following contributions:

- We carefully and comprehensively assess the progress of recently developed DLKT algorithms through the lens of empirical evaluation of critical experimental setup and design considerations on a variety of public datasets. Through the empirical studies, we attempt to provide answers to the following research questions:
  - **RQ1**: What is a reasonable, reliable and realistic evaluation process for DLKT algorithms?
  - **RQ2**: How do different characteristics of student data, model design and prediction scenario affect the model performance?
- To ensure reproducibility and foster future research, we develop PYKT, an easy-to-use and end-to-end PyTorch benchmark library that includes critical data preprocessing, standardized dataset splitting, SOTA DLKT implementations, and real-world evaluation protocols in educational contexts. We hope the use of PYKT will greatly relieve the burden of comparing existing baselines and developing new algorithms. The PYKT library is open-sourced at `https://pykt.org/`.

Please note that there have been several KT related survey papers [2, 17, 27, 28]. However, to the best of our knowledge, none of the existing work provides rigorous empirical evidence of DLKT approaches on the standardized experimental evaluation in real-world educational contexts.

# 2 Problem Statement

Let $\mathcal{Q}$, and $\mathcal{C}$ be the sets of questions, and KCs respectively. Let each student $\mathbf{s}$ be a chronologically ordered collection of historical learning interactions, i.e., $\mathbf{s} = \{\mathbf{e}_j\}_{j=1}^{n}$ where $\mathbf{e}_j$ is the $j$th interaction

---

[2]A KC is a generalization of everyday terms like concept, principle, fact, or skill.

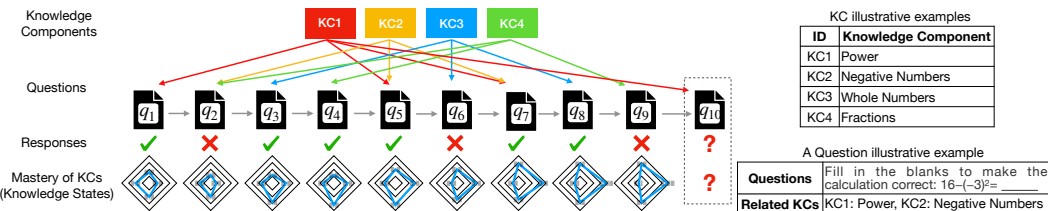

Figure 1: The graphical illustration of the KT problem.

and $n$ is the total number of interactions for $\mathbf{s}$. We denote an interaction $\mathbf{e}$ as a 4-tuple, i.e., $\mathbf{e} = <q, \{c\}, r, t>$, where $q, \{c\}, r, t$ represent the specific question, the associated KC set, the binary valued student response[3], and student's response timestamp respectively. As illustrated in Figure 1, in this work, our objective is to predict the values $(\hat{y}_{x_*})$ of mastery levels of questions or KCs for the target student $\mathbf{s}$ given his/her past learning interaction data where $x_*$ can be arbitrary questions or KCs, i.e., $x_* = q_*$ or $k_*$, where $q_* \in \mathcal{Q}$ and $k_* \in \mathcal{C}$.

Please note that in real-world educational contexts, the number of questions is substantially larger than the number of KCs. To avoid the over parameterization, the majority of existing DLKT approaches conduct mastery predictions of questions indirectly by aggregated predicted mastery levels of KCs [10]. When KC information is not available, mastery levels of questions can be directly modeled on question-response data.

## 3 The DLKT Benchmark

### 3.1 Representative DLKT Methods

To carefully assess the progress of DLKT model-wise and application-wise, we surveyed existing DLKT related publications in top AI/ML venues[4] from 2015-2021[5] and collected all the baselines compared in these research works and selected the most frequently mentioned DLKT baselines as our representative DLKT methods[6]. In addition, we include some recently proposed approaches to cover a very wide range of DLKT models with different design focus. Please note that there are a few newly developed DLKT focusing on either utilizing more auxiliary information such as LPKT that utilizes time spent on questions [29], or solving data isolation problems in KT via federated learning [42]. The generalization of these approaches are limited to specific datasets and are out of scope in this paper. These representative DLKT methods[7] can be categorized as follows:

- **Deep sequential models**: the chronologically ordered interaction sequence is captured by deep sequential models such as LSTM and GRU [4, 12, 15, 16, 20, 21, 25, 34, 45]. Selected approaches are **DKT** [25], **DKT+** [45], **DKT-F** [21], and **KQN** [15].

  - **DKT**: leverages an LSTM layer to encode the student knowledge state to predict the students' response performances [25].
  - **DKT+**: an improved version of DKT to solve the reconstruction and non-consistent prediction problems [45].
  - **DKT-F**: an extension of DKT that model the students' forgetting behaviors [21].
  - **KQN**: uses student knowledge state encoder and skill encoder to predict the student response performance via the dot product [15].

- **Memory augmented models**: the latent relations between KCs are explicitly modeled by an external memory that is updated iteratively [1, 29, 46]. Selected approach is **DKVMN** [46].

  - **DKVMN**: designs a static key matrix to store the relations between the different KCs and a dynamic value matrix to update the students' knowledge state [46].

---

[3]Student response is a binary valued indicator variable where 1 represents the student correctly answered the question, and 0 otherwise.

[4]Venues include NeurIPS, ICML, ICLR, AAAI, IJCAI, KDD, WWW, SIGIR, MM, WSDM, ICDM, CIKM.

[5]The very first deep knowledge tracing model is proposed by Piech et al. at NIPS 2015 [25].

[6]The comprehensive DLKT baseline frequency summary is listed in Appendix A.1.

[7]We also include detailed model explanation in the PYKT library docs at `https://pykt-toolkit.readthedocs.io/en/latest/models.html`.

- **Adversarial based models**: the adversarial training techniques such as adversarial perturbations are applied into the original student interaction sequence to reduce the risk of DLKT overfitting and limited generalization problem [12]. Selected approach is **ATKT** [12].
  - **ATKT**: performs adversarial perturbations into student interaction sequence to improve DLKT model's generalization ability [12].
- **Graph based models**: the response interactions between students and questions and the knowledge associations between questions and KCs form a tripartite graph and graph based techniques are applied to aggregate such relations [22, 37, 44]. Selected approach is **GKT** [22].
  - **GKT**: utilizes the graph structure to predict the student response performance [22].
- **Attention based models**: dependence between interactions is captured by the attention mechanism and its variants [10, 24, 26, 47]. Selected approaches are **AKT** [10], **SAKT** [23] and **SAINT** [7].
  - **AKT**: leverages an attention mechanism to characterize the time distance between questions and the past interaction of students [10].
  - **SAKT**: utilizes a self-attention mechanism to capture relations between exercises and the student responses [23].
  - **SAINT**: uses the Transformer-based encoder-decoder architecture to capture students' exercise and response sequences [7].

Please note that the above categorizations are not exclusive and related techniques can be combined. For example, Ghodai and Qing proposed a sequential key-value memory network to unify the strengths of recurrent modeling capacity and memory capacity [1].

## 3.2 Datasets

We select 7 widely used datasets to evaluate the performance of the popular models. The original raw data download links are listed in Appendix A.2. We will briefly introduce the details of each dataset and the data statistics are shown in Table 1.

- **Statics2011**: This dataset is collected from an engineering statics course taught at the Carnegie Mellon University during Fall 2011 [33]. Recommended by [6, 10, 46], a unique question is constructed by concatenating the problem name and step name.
- **ASSISTments2009**: This dataset is made up of math exercises, collected from the free online tutoring ASSISTments platform in the school year 2009-2010. The dataset is widely used and has been the standard benchmark for KT methods over the last decade [1, 9, 10, 23, 43, 46].
- **ASSISTments2015**: Similar to ASSISTments2009, this dataset is collected from the ASSISTments platform in the year of 2015. This dataset has the largest number of students among the other ASSISTments datasets.
- **Algebra2005**: This dataset is from the KDD Cup 2010 EDM Challenge that contains 13-14 year old students' responses to Algebra questions [32]. It contains detailed step-level student responses. The unique question construction is similar to the process used in Statics2011.
- **Bridge2006**: This dataset is also from the KDD Cup 2010 EDM Challenge and the unique question construction is similar to the process used in Statics2011.
- **NIPS34**: This dataset is from the Tasks 3 & 4 at the NeurIPS 2020 Education Challenge. It contains students' answers to multiple-choice diagnostic math questions and is collected from the Eedi platform [40]. For each question, we choose to use the leaf nodes from the subject tree as its KCs.
- **POJ**: This dataset consists of programming exercises and is collected from Peking coding practice online platform. The dataset is originally scraped by Pandey and Srivastava [24].

## 3.3 The Standardized Evaluation Protocol

A standardized evaluation protocol is one of the most important basis of AI research and it impacts model performance, fairness, and robustness. Even with many public KT datasets, due to the lack of agreed upon training and evaluation procedures, the published DLKT results surprisingly diverge. For example, the reported AUC scores of DKT and AKT on ASSISTments2009 range from 0.73

Table 1: Data statistics of 7 datasets in PYKT. "Original" and "After Preprocessing" refer to the initial and preprocessed data statistics. "avg KCs" denotes the number of average KCs per question.

| Datasets | Original | | | | | After Preprocessing | | | | |
|---|---|---|---|---|---|---|---|---|---|---|
| | interactions | sequences | questions | KCs | avg KCs | interactions | sequences | questions | KCs | avg KCs |
| **Statics2011** | 194,947 | 333 | 1,224 | - | - | 189,292 | 1,034 | 1,223 | - | - |
| **ASSISTments2009** | 346,860 | 4,217 | 26,688 | 123 | 1.1969 | 337,415 | 4,661 | 17,737 | 123 | 1.1970 |
| **ASSISTments2015** | 708,631 | 19,917 | - | 100 | - | 682,789 | 19,292 | - | 100 | - |
| **Algebra2005** | 809,694 | 574 | 210,710 | 112 | 1.3634 | 884,098 | 4,712 | 173,113 | 112 | 1.3634 |
| **Bridge2006** | 3,679,199 | 1,146 | 207,856 | 493 | 1.0136 | 1,824,310 | 9,680 | 129,263 | 493 | 1.0136 |
| **NIPS34** | 1,382,727 | 4,918 | 948 | 57 | 1.0148 | 1,399,470 | 9,401 | 948 | 57 | 1.0148 |
| **POJ** | 996,240 | 22,916 | 2,750 | - | - | 987,593 | 20,114 | 2,748 | - | - |

to 0.821 [20, 45] and from 0.747 to 0.835 in [10, 38] respectively. To this end, we standardize the prediction scenarios (Section 3.3.1), data preprocessing, training and testing sets splitting (Section 3.3.2) and evaluation metric (Section 3.3.3).

### 3.3.1 Real-world Prediction Scenarios

**Train DLKT models on KCs but evaluate them on questions**. In real-world educational scenarios, the question bank is usually much bigger than the set of KCs. For example, the number of questions is more than 1500 times larger than the number of KCs in Algebra2005 (see Table 1). Therefore, to effectively learn and fairly evaluate the DLKT models from such highly sparse question-response data, a recommended process[8] is listed as follows (also shown in Figure 2):

- **Step 1**: Train the DLKT models on KC-response data, which is artificially generated from question-response data by expanding each question-level interaction into multiple KC-level interactions when the question is associated with a set of KCs.

- **Step 2**: Use the learned DLKT models to predict on the above expanded KC-response data first and then output the final question-level predictions by aggregating predicted mastery levels of its KCs.

Although predictions at both question level and KC level are very important and useful for building personalized educational applications, when conducting offline model comparisons, it is recommended that the DLKT models are evaluated on prediction tasks at the question level instead of at the KC level. This is because (1) we only observe student responses on questions and have no ground truth about KCs; (2) a question may be associated with multiple KCs and evaluation results at the KC level may overestimate or underestimate the real model performance [41].

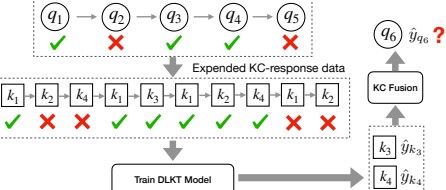

Figure 2: A recommended procedure for training and evaluating the DLKT models.

Specifically, implementing the above evaluation procedure of "step 2" in practice is worthy of serious attention: *predictions of mastery level of KCs within a question should be predicted in an "all-in-one" manner*. The "all-in-one" prediction approach requires to simultaneously estimate the mastery level of all the KCs under each specific question. As illustrated in Figure 2, when predicting the outcome of $q_6$ that is associated with both $k_3$ and $k_4$, we should estimate $\hat{y}_{k_3}$ and $\hat{y}_{k_4}$ independently at the same time. Surprisingly, this crucial issue is neglected in some existing works, whose open sourced implementations conduct one-by-one evaluation on the expanded KC-response sequence, i.e., predicting $\hat{y}_{k_{t+1}}$ given all the responses (or labels) of $\hat{y}_{k_1}, \hat{y}_{k_2}, \cdots, \hat{y}_{k_t}$ are known. Unfortunately, this will cause the leakage of the ground truth since consecutive KCs like $k_t$ and $k_{t+1}$ may be associated with the same questions, which is referred to as the **label leakage** problem. Such dependent predictions will artificially boost prediction performance [41] and empirical analysis on this issue is discussed in Section 4.1.

**KC prediction aggregation**. To conduct a fine-grained, comprehensive and fair model comparison, we consider 4 different ways of aggregating the KC predictions in the above recommended process (depicted in the "KC Fusion" module in Figure 2), which include (1) **early fusion** that uses the averaged hidden states of all associated KCs to predict the question-level response, i.e., *EF*; (2) **late fusion - average** that uses the averaged prediction probabilities of all KCs as the question-level prediction probability, i.e., *LF-AVG*; (3) **late fusion - majority vote** that conducts majority votes based on all KC prediction results, i.e., *LF-MV*; (4) **late fusion - strict** that predicts positive if and only if all the related KCs' predicted labels are positive, i.e., *LF-S*.

---

[8]The below procedure is also briefly mentioned by Ghosh et al. [10].

**One-step and multi-step ahead KT predictions**. To make the benchmark close to the real application scenarios, we divide our prediction scenarios into two settings: (1) one-step ahead prediction; and (2) multi-step ahead prediction. Specifically, the one-step ahead prediction task only predicts the student's response on the last question given the student's historical interaction sequence (depicted in Figure 3(a)). While the multi-step ahead prediction task predicts a span of student's responses given the student's historical interaction sequence (depicted in Figure 3(b)). Accurate one-step ahead prediction will largely improve the real-time educational recommender systems and the multi-step ahead prediction will provide constructive feedback to learning path selection and construction and help teachers adaptively adjust future teaching materials as well.

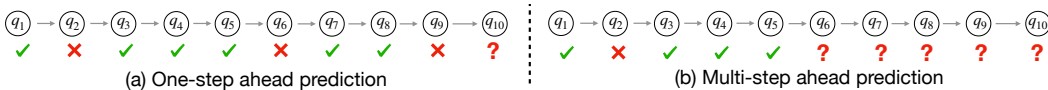

Figure 3: Two different prediction scenarios: one-step ahead and multi-step ahead predictions.

### 3.3.2 Data Preprocessing

Another challenge of reproducing existing DLKT research is the lack of standardized data preprocessing procedure. Aforementioned public KT datasets are far away from the ready-to-use stage and require many data preprocessing steps such as removing duplication, handling null or invalid values, and re-constructing the chronological interaction order. Furthermore, such steps are rarely described in existing publications and many open-sourced packages do not include data scripts. Therefore, in this work, our PYKT benchmark aims to outline a reasonable data preprocessing procedure for DLKT research. Specifically, the procedure includes: (1) data filtering: filter out interactions if no student id or any type of information of our 4-tuple interaction representation is not available or missing and filter out students if their sequences have less than 3 interactions. (2) data splitting: 20% students (their interaction sequences) are randomly withheld as the test set. The rest 80% students are randomly and evenly split into 5 folds: 4 folds for training and 1 fold for validation. (3) KC subsequence generation for training and validation. Expand original question-response sequence into KC level by repeating responses multiple times when a question has more than one KCs, one for each KC. Truncate the expanded KC level response sequence into shorter subsequences of length $m$, where $m$ is the pre-defined max training sequence length. Sequences or subsequences whose length is less than $m$ will be padded with -1. The data statistics of the aforementioned 7 datasets after preprocessing are shown in Table 1.

### 3.3.3 Evaluation Metric

Similar to all existing DLKT research works [1, 4, 10, 12, 15, 16, 20, 21, 22, 24, 25, 26, 29, 34, 37, 44, 45, 46, 47], we use the area under the receiver operating characteristics curve (AUC) as the main metric to evaluate the performance of DLKT models on predicting binary-valued future learner responses to either questions or KCs. Meanwhile, we also include detailed model performance in terms of accuracy in Appendix.

## 4 Empirical Studies

Let AS2009, AS2015, AL2005, BD2006 represent datasets ASSISTments2009, ASSISTments2015, Algebra2005, and Bridge2006 respectively in the following section.

**Experimental setup**. Since we have both question related and KC related information available for datasets of AS2009, AL2005, BD2006, and NIPS34, we conduct offline experiments following the recommended procedure discussed in Section 3.3.1. For experiments on datasets such as Statics2011, AS2015, and POJ that question related or KC related information is missing, both the training and testing procedures are conducted on question-response data. The details of training, validation and test sets are described in Section 3.3.2. By default, we choose to use LF-AVG for KC prediction fusion and all experiments are conducted in the one-step ahead prediction setting and the detailed analysis of multi-step ahead prediction is discussed in Observation 5 in Section 4.1. The pre-defined max training sequence length $m$ is set to 200.

**Implementation details**. We use the Bayesian search method to find the best hyperparameter and stop searching when the number of the tuned hyperparameter combinations in each data fold is larger than

200 and there is no AUC improvement in the last 50 rounds. For each hyperparameter combination, we adopt the Adam optimizer to train all models with the early stop heuristic. We compute the AUC score on the development set for every epoch and update the best AUC score if needed. We stop the training process when the AUC score doesn't improve in the last 10 iterations or the number of training epochs reaches 200. Experiments are conducted at `https://wandb.ai/`. Please note that all the following AUC results are slightly different (less than 0.005) compared to the results in the original version since we re-run all the experiments under a more consistent hyperparameter search space. Detailed can be found in Appendix A.3.

## 4.1 Results

We provide extensive experiments and analyses to demonstrate the value of PYKT benchmark. Insights, findings and suggestions are summarized as observations described as follows.

**Observation 1. Attention mechanism greatly affects DLKT model performance. The DKT model that first applies deep learning into the KT problem is still superior.**

Table 2 summarizes the overall AUC performance results. Specifically, for datasets, i.e., AS2009, AL2005, BD2006, and NIPS34 in which both question and KC related information is available, the DLKT model performance is evaluated on both the original observed question-response data and the expended KC-response data. We have performed 5-fold cross validation for all of our experiments and the averaged AUC score is reported. The score standard deviations and the accuracy scores are reported in Appendix A.4 due to space limit. From Table 2, we find the following results: (1) AKT outperforms (almost) all other methods on all datasets. Since AKT proposes a monotonic attention mechanism to capture short-term dependencies on the past at different time scales, it outperforms the standard self-attention based baselines like SAKT and SAINT. Furthermore, AKT learns the Rasch model-based embeddings that implicitly model question difficulties when both question and KC information is available. This significantly improves its prediction performance and AKT is 3.06%, 1.50%, 1.62% and 1.60% better than the second best approach on AS2009, AL2005, BD2006 and NIPS34 datasets. Meanwhile, it only exceeds the second best approach by 0.30% and 1.08% on Statics2011 and POJ that either question information or KC information is missing. (2) The majority of recently proposed DLKT approaches cannot beat the vanilla DKT model and the performance between DKT and the best performing model for each dataset is shown in the bottom line ($\Delta_{\mathbf{DKT}}$) in Table 2. (3) Deep sequential models, i.e., DKT, DKT+, DKT-F, KQN, outperform the self-attention based model SAKT and SAINT in majority cases. We believe this is because different from the most NLP tasks that require capturing the long-term dependencies, faraway historical interactions have little influence on students' future performance prediction. This also indicates that the recency and forgetting effects need to be considered in the DLKT model design. (4) KT prediction on programming exercises is much harder compared to KT tasks on math questions. DLKT models are able to achieve 0.8ish AUC scores but only get 0.6ish AUC on the POJ dataset. (5) There is little difference in prediction results on BD2006 and NIPS34 datasets at question level and KC level. This is because most questions in these two datasets only contain one KC and the averaged numbers of KCs per question are 1.0136 and 1.0148 in BD2006 and NIPS34 respectively. (6) The original ATKT implementation utilizes the future ground truth to predict the mastery level of the current question, which is problematic. The published promising results cannot be reproduced when this problem gets fixed. The fixed results are reported in this paper and details are discussed in Appendix A.5.

Table 2: The overall prediction performance in terms of AUC at both question level and KC level. Marker ∗, ○ and ● indicates whether the AKT model is statistically superior/equal/inferior to the compared method (using paired t-test at 0.01 significance level). The last column shows the total numbers of win/tie/loss for AKT against the compared method (e.g., #win is how many times AKT significantly outperforms that method).

| Model | Question Level(All-in-One) | | | | KC Level(ALL-in-One) | | | | Statics2011 | AS2015 | POJ | AKT #win/#tie/#loss |
|---|---|---|---|---|---|---|---|---|---|---|---|---|
| | AS2009 | AL2005 | BD2006 | NIPS34 | AS2009 | AL2005 | BD2006 | NIPS34 | | | | |
| **DKT** | 0.7541* | 0.8149* | 0.8015* | 0.7689* | 0.7419* | 0.8146● | 0.8013* | 0.7681* | 0.8222* | 0.7271* | 0.6089* | 10/0/1 |
| **DKT+** | 0.7547* | 0.8156* | 0.8020* | 0.7696* | 0.7424* | 0.8144● | 0.8019* | 0.7689* | 0.8279* | 0.7285● | 0.6173* | 9/0/2 |
| **DKT-F** | - | 0.8147* | 0.7985* | 0.7733* | - | 0.8163● | 0.7984* | 0.7727* | 0.7839* | - | 0.6030* | 7/0/1 |
| **KQN** | 0.7477* | 0.8027* | 0.7936* | 0.7684* | 0.7361* | 0.8005* | 0.7935* | 0.7677* | 0.8232* | 0.7254* | 0.6080* | 11/0/0 |
| **DKVMN** | 0.7473* | 0.8054○ | 0.7983* | 0.7673* | 0.7330* | 0.7891* | 0.7981* | 0.7668* | 0.8093* | 0.7227* | 0.6056* | 10/1/0 |
| **ATKT** | 0.7470* | 0.7995* | 0.7889* | 0.7665* | 0.7337* | 0.7964* | 0.7885* | 0.7658* | 0.8055* | 0.7245* | 0.6075* | 11/0/0 |
| **GKT** | 0.7424* | 0.8110* | 0.8046* | 0.7689* | 0.7227* | 0.8025* | 0.8045* | 0.7681* | 0.8040* | 0.7258* | 0.6070* | 11/0/0 |
| **SAKT** | 0.7246* | 0.7880* | 0.7740* | 0.7517* | 0.7085○ | 0.7682* | 0.7738* | 0.7516* | 0.7965* | 0.7114* | 0.6095* | 10/1/0 |
| **SAINT** | 0.6958* | 0.7775* | 0.7781* | 0.7873* | 0.6865* | 0.6662* | 0.7779* | 0.7860* | 0.7599○ | 0.7026* | 0.5563* | 10/1/0 |
| **AKT** | 0.7853 | 0.8306 | 0.8208 | 0.8033 | 0.7650 | 0.8091 | 0.8206 | 0.8017 | 0.8309 | 0.7281 | 0.6281 | - |
| $\Delta_{\mathbf{DKT}}$ | 0.0312 | 0.0157 | 0.0193 | 0.0344 | 0.0231 | 0.0017 | 0.0193 | 0.0336 | 0.0087 | 0.0014 | 0.0192 | - |

**Observation 2. One-by-one evaluation on expanded KC sequences causes label leakage problem that leads to performance inflation.**

As discussed in Section 3.3.1, label leakage happens when the target KC prediction $\hat{y}_{k_{j+1}}$ depends on the ground truth of the previous KC $k_j$ in the expanded KC sequence and at the same time, $k_j$ and $k_{j+1}$ belong to the same question. During our journey of building PYKT, we found many publicly available DLKT implementations overlook this leakage issue [10, 23, 46], which artificially

Table 3: The boosted DLKT AUC results due to label leakage.
The exaggerated gains ($\triangle_{\text{Gain}}$) are computed by subtracting AUC scores of one-by-one predictions (left part in Table 3) from AUC scores of all-in-one predictions at KC level (middle part in Table 2).

| Model | KC Level(One-by-One) | | | | Exaggerated Performance Gains ($\triangle_{\text{Gain}}$) | | | |
|---|---|---|---|---|---|---|---|---|
| | AS2009 | AL2005 | BD2006 | NIPS34 | AS2009 | AL2005 | BD2006 | NIPS34 |
| **DKT** | 0.8262 | 0.9218 | 0.8028 | 0.7742 | 0.0843 | 0.1072 | 0.0015 | 0.0061 |
| **DKT+** | 0.8268 | 0.9221 | 0.8032 | 0.7748 | 0.0844 | 0.1077 | 0.0013 | 0.0059 |
| **DKT-F** | - | 0.9220 | 0.7997 | 0.7787 | - | 0.1057 | 0.0013 | 0.0060 |
| **KQN** | 0.8216 | 0.9179 | 0.7949 | 0.7736 | 0.0855 | 0.1174 | 0.0014 | 0.0059 |
| **DKVMN** | 0.8213 | 0.9190 | 0.7993 | 0.7723 | 0.0883 | 0.1299 | 0.0012 | 0.0055 |
| **ATKT** | 0.8210 | 0.9156 | 0.7902 | 0.7718 | 0.0873 | 0.1192 | 0.0017 | 0.0060 |
| **GKT** | 0.8171 | 0.9208 | 0.8057 | 0.7741 | 0.0944 | 0.1183 | 0.0012 | 0.0060 |
| **SAKT** | 0.7806 | 0.9115 | 0.7740 | 0.7532 | 0.0721 | 0.1433 | 0.0002 | 0.0016 |
| **SAINT** | 0.7605 | 0.9050 | 0.7787 | 0.7910 | 0.0740 | 0.2388 | 0.0008 | 0.0050 |
| **AKT** | 0.8493 | 0.9305 | 0.8218 | 0.8084 | 0.0843 | 0.1214 | 0.0012 | 0.0067 |

boosts prediction performance. We reproduce such "wrong" evaluation procedure and report the exaggerated results in the left part in Table 3. Furthermore, we explicitly show the exaggerated AUC gains ($\triangle_{\text{Gain}}$) in the right part of Table 3 by computing the AUC scores difference between results of one-by-one predictions at KC level (values in Table 3) and results of all-in-one predictions at KC level (middle part in Table 2). The related accuracy results are reported in Appendix A.6. As we can see that, the leakage issue is much worse in AS2009 and AL2005, i.e., the mean values of $\triangle_{\text{Gain}}$ is 8.38% and 13.09% respectively. This is because questions in AS2009 and AL2005 have more KCs associated and their average KC numbers per question is 1.1970 and 1.3634 compared to the value of 1.0136 and 1.0148 in BD2006 and NIPS34. Experimental results and conclusions impacted by the leakage problem need to be re-validate and we believe this is the reason why we cannot reproduce the results of many DKLT models on AS2009 and AL2005 [10, 23, 46]. In summary, we do not recommend to conduct KT evaluations at KC level in the above one-by-one manner.

**Observation 3. DLKT models behaves differently for students who have very long interaction sequences.**

In the real educational scenarios, the length of historical learning interactions varies a lot. Therefore, we split the test set into two parts: (1) students with long interaction sequences (sequence length is larger than 200), denoted as $L$; and (2) students with short interaction sequences (sequence length is less than or equal to 200), denoted as $S$. We choose 200 as the cutoff threshold because the pre-defined max training sequence length $m$ is set to 200 as well in all experiments. We evaluate the DLKT performance on $L$ and $S$ respectively (shown in Table 4). Its full version with standard deviations and the results in terms of accuracy are reported in Appendix A.7 and Appendix A.8. Generally speaking, DLKT models perform quite differently on $L$ and $S$. For example, the difference of AUC scores on POJ ranges from 5% to 14%. Meanwhile, DLKT models perform alike on Statics2011 and AS2015. By digging into these two datasets, we find that these two datasets have very high numbers of KC co-occurrence on $L$ and $S$, i.e., the appearance of one KC is often accompanied by another KC. Such adjacent interactions will largely influence the model performance and the role of long-term contextual information reduces quite a lot. As a result, there are no obvious AUC performance difference on $L$ and $S$ student subgroups.

Table 4: Prediction performance in terms of AUC for students with different lengths of interactions.

| Model | Statics2011 | | AS2009 | | AS2015 | | AL2005 | | BD2006 | | NIPS34 | | POJ | |
|---|---|---|---|---|---|---|---|---|---|---|---|---|---|---|
| | L | S | L | S | L | S | L | S | L | S | L | S | L | S |
| **DKT** | 0.8219 | 0.8314 | 0.7351 | 0.7650 | 0.7106 | 0.7281 | 0.8160 | 0.7623 | 0.8010 | 0.8563 | 0.7740 | 0.7430 | 0.5979 | 0.6629 |
| **DKT+** | 0.8276 | 0.8364 | 0.7357 | 0.7657 | 0.7113 | 0.7296 | 0.8168 | 0.7600 | 0.8015 | 0.8593 | 0.7748 | 0.7436 | 0.6045 | 0.6782 |
| **DKT-F** | 0.7859 | 0.7465 | - | - | - | - | 0.8158 | 0.7597 | 0.7980 | 0.8467 | 0.7784 | 0.7480 | 0.5915 | 0.6606 |
| **KQN** | 0.8230 | 0.8280 | 0.7259 | 0.7604 | 0.7064 | 0.7266 | 0.8038 | 0.7466 | 0.7931 | 0.8515 | 0.7738 | 0.7414 | 0.5944 | 0.6774 |
| **DKVMN** | 0.8086 | 0.8294 | 0.7271 | 0.7588 | 0.7039 | 0.7240 | 0.8067 | 0.7429 | 0.7978 | 0.8540 | 0.7725 | 0.7414 | 0.5924 | 0.6732 |
| **ATKT** | 0.8046 | 0.8295 | 0.7249 | 0.7605 | 0.7029 | 0.7262 | 0.8004 | 0.7564 | 0.7884 | 0.8464 | 0.7711 | 0.7438 | 0.5960 | 0.6687 |
| **GKT** | 0.8044 | 0.8004 | 0.7224 | 0.7535 | 0.7111 | 0.7266 | 0.8122 | 0.7528 | 0.8042 | 0.8535 | 0.7741 | 0.7431 | 0.5977 | 0.6577 |
| **SAKT** | 0.7958 | 0.8179 | 0.6989 | 0.7403 | 0.6857 | 0.7134 | 0.7891 | 0.7347 | 0.7734 | 0.8239 | 0.7570 | 0.7253 | 0.6001 | 0.6544 |
| **SAINT** | 0.7592 | 0.7845 | 0.6687 | 0.7112 | 0.6617 | 0.7060 | 0.7788 | 0.7097 | 0.7776 | 0.8189 | 0.7912 | 0.7687 | 0.5294 | 0.6702 |
| **AKT** | 0.8305 | 0.8466 | 0.7781 | 0.7878 | 0.7113 | 0.7292 | 0.8317 | 0.7771 | 0.8204 | 0.8643 | 0.8074 | 0.7829 | 0.6137 | 0.6949 |

**Observation 4. Prediction results of different ways of KC aggregation resemble each other and the "late fusion - average" is slightly better compared to other approaches.**

Generally speaking, there are 4 different ways of aggregating KC predictions in the "KC Fusion" module in Figure 2. Therefore, we conduct extensive experiments to empirically evaluate their impacts on the final prediction performance (shown in Table 5). Due to the space limit, we choose to use DKT and AKT as the representative approaches from the deep sequential DLKT models and attention based DLKT models. The full results of all the baselines are shown in Appendix A.9. Please note that since DKT, ATKT and GKT inherently don't use individual hidden states to model each KC,

EF approach is inapplicable on these methods. Similar to Table 2, we use markers ∗, ∘ and ● to indicate whether the **LF-AVG** model is significantly better/equal to/worse than the compared method at 0.01 significance level. The difference ($\Delta_{Fusion}$) is computed by subtracting AUC scores of LF-AVG from the best AUC score from LF-MV, LF-S, and EF. The last row shows the total number of win/tie/loss for LF-AVG against the compared method. As we can see, (1) though statistically significant, the performance difference between different fusion mechanisms is very small. The LF-AVG approach outperforms or perform alike other approaches across all 4 datasets. (2) $\Delta_{Fusion}$ is much larger on AL2005, we believe this is because the

Table 5: Impact on different KC fusion mechanisms.
Marker ∗, ∘ and ● indicates whether the LF-AVG mechanism is statistically superior/equal/inferior to the compared fusion method (using paired t-test at 0.01 significance level).

| Model | Dataset | Fusion Mechanisms | | | | $\Delta_{Fusion}$ |
| | | LF-AVG | LF-MV | LF-S | EF | |
|---|---|---|---|---|---|---|
| DKT | AS2009 | 0.7541 | 0.7526* | 0.7524* | - | 0.0015 |
| | AL2005 | 0.8149 | 0.8123* | 0.8131* | - | 0.0018 |
| | BD2006 | 0.8015 | 0.8015∘ | 0.8015∘ | - | 0.0000 |
| | NIPS34 | 0.7689 | 0.7687* | 0.7688* | - | 0.0001 |
| DKVMN | AS2009 | 0.7473 | 0.7458* | 0.7456* | 0.7454* | 0.0015 |
| | AL2005 | 0.8054 | 0.8022* | 0.8021* | 0.7961* | 0.0032 |
| | BD2006 | 0.7983 | 0.7983∘ | 0.7983∘ | 0.7983∘ | 0.0000 |
| | NIPS34 | 0.7673 | 0.7672* | 0.7673∘ | 0.7673∘ | 0.0000 |
| ATKT | AS2009 | 0.7470 | 0.7440* | 0.7466* | - | 0.0004 |
| | AL2005 | 0.7995 | 0.7963* | 0.7974* | - | 0.0021 |
| | BD2006 | 0.7889 | 0.7888* | 0.7889∘ | - | 0.0000 |
| | NIPS34 | 0.7665 | 0.7663* | 0.7665∘ | - | 0.0000 |
| GKT | AS2009 | 0.7424 | 0.7376* | 0.7401* | - | 0.0023 |
| | AL2005 | 0.8110 | 0.8072* | 0.8072* | - | 0.0038 |
| | BD2006 | 0.8046 | 0.8046∘ | 0.8046∘ | - | 0.0000 |
| | NIPS34 | 0.7689 | 0.7686* | 0.7689∘ | - | 0.0000 |
| AKT | AS2009 | 0.7853 | 0.7794* | 0.7847* | 0.7825* | 0.0006 |
| | AL2005 | 0.8306 | 0.8228* | 0.8275* | 0.8177* | 0.0031 |
| | BD2006 | 0.8208 | 0.8208∘ | 0.8208∘ | 0.8208∘ | 0.0000 |
| | NIPS34 | 0.8033 | 0.8028* | 0.8033∘ | 0.8034● | -0.0001 |
| #win/#tie/#loss | | - | 16/4/0 | 11/9/0 | 4/3/1 | |

AL2005 dataset has the largest the number of KCs per question (shown in Table 2). Averaging probabilities of KCs within a question will stabilize the DLKT model performance.

**Observation 5. The choice of accumulative or non-accumulative prediction vastly influences the DLKT performance in the multi-step ahead prediction scenario.**

As discussed in Section 3.3.1, accurate predictions in the multi-step ahead prediction scenario are also very important from educational perspectives. Practically, there are two different approaches, i.e., accumulative prediction and non-accumulative prediction. The accumulative prediction approach uses the last predicted values for the current prediction while the non-accumulative prediction predicts all future values all at once. To have a fine-grained analysis in the multi-step ahead prediction scenario, we further experiment with DLKT models on different portions of observed student interactions. Specifically, we vary the observed percentages of student interaction length from 20% to 90% with step size of 10%. Due to the space limit, we select DKT/DKVMN/ATKT/GKT/AKT and AS2009/BD2006/POJ as the representative approaches and datasets and the results are shown in Figure 4. The full AUC and accuracy results are shown in Appendix A.10 and Appendix A.11.

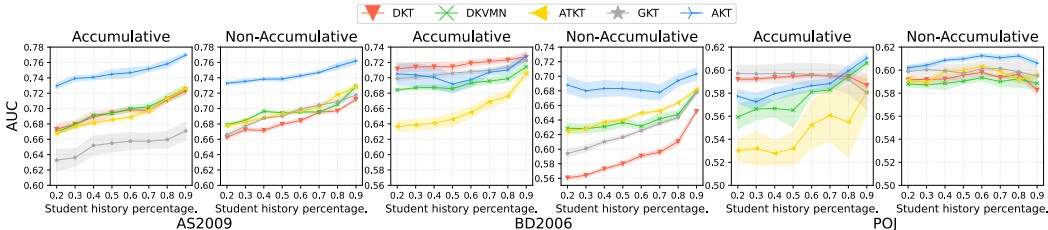

Figure 4: Accumulative and non-accumulative predictions in the multi-step ahead scenario in terms of AUC on AS2009, BD2006, and POJ.

We make the following observations: (1) accumulative performance is mostly worse than non-accumulative performance on the POJ dataset. This is because the DLKT approaches only achieve 0.6ish AUC (shown in Table 2) and the previous prediction errors are aggregated and propagated to the future prediction tasks in an accumulative fashion. (2) the performance of non-accumulative predictions on BD2006 drops a lot compared to its accumulative prediction results. We believe this is because the BD2006's has the largest average sequence length. When conducting non-accumulative predictions with different percentages of student historical interactions, it actually needs to predict questions in the very farway future all in once. The DLKT performance downgrades when there is not enough contextual information of the target question. (3) with the increase of student historical interactions, the DLKT performance gradually improves in both the accumulative and non-accumulative settings.

# 5  Limitation

While PYKT is able to standardize and accelerate research in DLKT, we are aware of some potential limitations described as follows:

- **Binary response assumption**. Most existing DLKT research relies on the binary response assumption and existing publicly available datasets only contain the binary labels, i.e., correct or incorrect. We believe this is because the binary valued data is (sort of) the most objective assessment labels that can be easily collected from either the online learning platforms or the offline classrooms in a large scale. However, we deeply believe that having more fine-grained student assessment labels beyond binary value of correct and incorrect would definitely push the boundary of this research field. Some research works such as option tracing [11] aim to extend existing KT methods beyond correctness prediction to the task of predicting the exact option students select in multiple choice questions. This only works for multiple choice questions and it generalizes the traditional KT prediction from binary classification to the multi-class classification settings. The PYKT library is well modularized and different loss functions or evaluation metrics can be easily added. Hence, it is flexible to extend the existing benchmark to non-binary prediction settings such as multi-class classification or real-valued regression problems.
- **Auxiliary side information**. All existing baselines don't utilize the rich auxiliary side information in educational contexts. Various auxiliary side information could be extracted as external knowledge and integrated with the DLKT models. Such auxiliary knowledge is expected to improve DLKT performance, which can be considered as follows:
  - **Question side info.**: (1) question text content [16, 34]; (2) latent question variations with respect to each KC [10]; (3) question difficulty level [10, 18, 47]; and (4) relations among questions [18, 24, 47].
  - **Student side info.**: (1) historical successful and failed attempts [47]; (2) recent attempts [47]; (3) students' learning ability [20]; and (4) individualized prior knowledge of students [30].
  - **KC side info.**: (1) latent knowledge representation [10, 15]; and (2) relations among KCs [18].

# 6  Conclusion

**Outlook**: We describe concrete ongoing and future work towards expanding PYKT. Specifically,

- We will keep adding newly developed DLKT approaches into our DLKT model zoo and provide ready-to-use benchmark results for both researchers and practitioners. We will incorporate more diverse datasets into the PYKT platform.
- We will explore new DLKT opportunities such as modeling external side information from the student side, KC side and question side. For example, we would like to apply the successful pre-training techniques into the KT domain to better capture the heterogeneous educational data.

**Conclusion**: We present PYKT, a systematic and comprehensive python library standardizing previous efforts in DLKT research with a focus on accessibility, reproducibility and practical usage in real-world educational contexts, thereby paving the way towards a deeper understanding of different DLKT components. Through its modularized and standardized data preprocessing components, rigorous and real-world prediction scenario formulation, and experiment management, PYKT is able to greatly relieve the burden of comparing existing baselines and developing new algorithms and highlights several future directions in building more practical, generalizable, and robust DLKT models.

# Acknowledgments

This work was supported in part by National Key R&D Program of China, under Grant No. 2020AAA0104500; in part by Beijing Nova Program (Z201100006820068) from Beijing Municipal Science & Technology Commission; in part by NFSC under Grant No. 61877029 and in part by Key Laboratory of Smart Education of Guangdong Higher Education Institutes, Jinan University (2022LSYS003).

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
