# A Appendix

## A.1 Frequency Summary of Representative DLKT Baselines in Top AI/ML Venues from 2015-2021

To systematically evaluate various neural architectures proposed in top AI/ML venues including NeurIPS, ICML, ICLR, AAAI, IJCAI, KDD, WWW, SIGIR, MM, WSDM, ICDM, and CIKM from 2015-2021, as seen in Table 6. We collect all the baselines compared in these research works and select the top 3 most frequently mentioned DLKT baselines as our representative DLKT methods.

Table 6: Frequency summary of DLKT baselines in top AI/ML venues from 2015-2021.

| Conference | Model | DKT | DKVMN | SAKT | AKT | DKT+ | GKT | EERNN | CKT | DHKT | DKT-F | PEBG | GIKT | EKT | DFKT |
|---|---|---|---|---|---|---|---|---|---|---|---|---|---|---|---|
| NIPS 2015 | DKT[25] | | | | | | | | | | | | | | |
| WWW 2017 | DKVMN[46] | ✓ | | | | | | | | | | | | | |
| AAAI 2018 | EERNN[34] | ✓ | | | | | | | | | | | | | |
| ICDM 2018 | DKT-DSC[20] | ✓ | | | | | | | | | | | | | |
| ICDM 2018 | PDKT-C[4] | ✓ | | | | | | | | | | | | | |
| WWW 2019 | DKT-F[21] | ✓ | | | | | | | | | | | | | |
| SIGIR 2019 | SKVMN[1] | ✓ | ✓ | | | | | | | | | | | | |
| CIKM2019 | DIRT[5] | | | | | | | | | | | | | | |
| AAAI 2020 | NeuralCD[39] | | | | | | | | | | | | | | |
| IJCAI 2020 | PEBG[18] | ✓ | ✓ | | | | | | | ✓ | | | | | |
| KDD 2020 | AKT[10] | ✓ | ✓ | ✓ | | ✓ | | | | | | | | | |
| SIGIR 2020 | CKT[30] | ✓ | ✓ | | | | | | | | | | | | |
| ICDM 2020 | SKT[37] | ✓ | ✓ | | | ✓ | ✓ | | | | | | | | |
| CIKM 2020 | RKT[24] | ✓ | ✓ | ✓ | | | | ✓ | | | ✓ | | | ✓ | |
| KDD 2021 | LPKT[29] | ✓ | ✓ | ✓ | ✓ | ✓ | | | ✓ | | | | | | |
| CIKM 2021 | MF-DAKT[47] | ✓ | ✓ | | ✓ | | ✓ | | | | | ✓ | ✓ | | |
| SIGIR 2021 | IEKT[19] | ✓ | ✓ | ✓ | ✓ | | ✓ | ✓ | ✓ | ✓ | | | | | ✓ |
| MM 2021 | ATKT[12] | ✓ | ✓ | ✓ | ✓ | ✓ | | | | | | | | | |
| WSDM 2021 | HawkesKT[38] | ✓ | | ✓ | ✓ | | | | | | ✓ | | | | |
| WSDM 2021 | FDKT[42] | ✓ | | | | | | | | | | | | | |
| | Total | 17 | 10 | 6 | 5 | 4 | 3 | 2 | 2 | 2 | 2 | 1 | 1 | 1 | 1 |

## A.2 Public Access of 7 KT Datasets

The original raw data used in this paper can be download as follows:

- **Statics2011**: `https://pslcdatashop.web.cmu.edu/DatasetInfo?datasetId=507`
- **AS2009**: `https://sites.google.com/site/assistmentsdata/home/2009-2010-assistment-data/skill-builder-data-2009-2010`
- **AS2015**: `https://sites.google.com/site/assistmentsdata/datasets/2015-assistments-skill-builder-data`
- **AL2005 & BD2006**: `https://pslcdatashop.web.cmu.edu/KDDCup/`
- **NIPS34**: `https://eedi.com/projects/neurips-education-challenge`
- **POJ**: `https://drive.google.com/drive/folders/1LRljqWfODwTYRMPw6wEJ_mMt1KZ4xBDk`

## A.3 Hyperparameter Search Details of Representative DLKT Baselines

As mentioned above in Section 4, we adopt Bayesian search method to find optimal hyperparameters for each model. We define a hyperparameter search space of all the DLKT models that considers experimental setups of the cited references [7, 10, 12, 15, 21, 22, 23, 25, 45, 46]. The detailed of the search spaces utilized in our DLKT models are listed in Table 7. Please note that due to the memory limitation of GPU devices, we set the hidden size of GKT as 16 on POJ and Statics2011 datasets.

## A.4 Overall *AUC* and *Accuracy* Results (with Standard Deviations) of 10 Representative DLKT Methods on 7 Datasets

Tables 8 - 10 summarize the overall AUC and accuracy performance results for all the representative baselines on 7 datasets across domains including mathematics and programming. Specifically, for datasets, i.e., AS2009, AL2005, BD2006, and NIPS34 in which both question and KC related

Table 7: Overview of hyperparameters used for all the models of PYKT benchmark. $\Upsilon, \Omega, \Lambda_1, \Lambda_2, \Lambda_3, \Phi, \Theta$ and $\Xi$ denote {1, 2, 4}, {4, 8}, {0, 0.05, 0.1, 0.15, 0.2, 0.25}, {0, 0.01, 0.03, 0.1, 0.3, 1}, {0, 0.3, 1, 3, 10, 30, 100}, {$10^{-3}, 10^{-4}, 10^{-5}$}, {0.05, 0.1, 0.3, 0.5} and {42, 3407} respectively. ★ denotes the hyperparameters are not tuned individually, they are consistent with their corresponding KC/Interaction/Key embedding size (e.g., the value embedding size of DKVMN is consistent with the key embedding size).

| | DKT | DKT+ | DKT-F | KQN | DKVMN | ATKT | GKT | SAKT | SAINT | AKT |
|---|---|---|---|---|---|---|---|---|---|---|
| Question/KC embedding size | - | - | - | {64,256} | - | {64,256} | - | - | {64,256} | {64,256} |
| Response embedding size | - | - | - | - | - | {64,256} | - | - | - | ★ |
| Interaction embedding size | {64,256} | {64,256} | {64,256} | {64,256} | {64,256} | - | {16,64} | {64,256} | - | - |
| Attention size | - | - | - | - | - | {64,256} | - | ★ | ★ | ★ |
| RNN/FFN layer size | ★ | ★ | ★ | {64,256} | - | {64,256} | ★ | ★ | ★ | {64,256} |
| Key embedding size | - | - | - | - | {32,64} | - | - | - | - | - |
| Value embedding size | - | - | - | - | ★ | - | - | - | - | - |
| Number of RNN layer | 1 | 1 | 1 | {1,2} | - | 1 | - | - | - | - |
| Number of attention blocks | - | - | - | - | - | - | - | $\Upsilon$ | $\Upsilon$ | $\Upsilon$ |
| Number of attention heads | - | - | - | - | - | - | - | $\Omega$ | $\Omega$ | $\Omega$ |
| Regularization $\lambda_r$ | - | $\Lambda_1$ | - | - | - | - | - | - | - | - |
| Regularization $\lambda_{w_1}$ | - | $\Lambda_2$ | - | - | - | - | - | - | - | - |
| Regularization $\lambda_{w_2}$ | - | $\Lambda_3$ | - | - | - | - | - | - | - | - |
| Perturbation rate $\epsilon$ | - | - | - | - | - | {1, 5, 10, 12, 15} | - | - | - | - |
| Adversarial loss weight $\beta$ | - | - | - | - | - | {0, 0.2, 0.5, 1, 2} | - | - | - | - |
| Batch size | 256 | 256 | 256 | 256 | 64 | 64 | 16 | 64 | 64 | 64 |
| Learning rate | $\Phi$ | $\Phi$ | $\Phi$ | $\Phi$ | $\Phi$ | $\Phi$ | $\Phi$ | $\Phi$ | $\Phi$ | $\Phi$ |
| Dropout rate | $\Theta$ | $\Theta$ | $\Theta$ | $\Theta$ | $\Theta$ | $\Theta$ | $\Theta$ | $\Theta$ | $\Theta$ | $\Theta$ |
| Random seed | $\Xi$ | $\Xi$ | $\Xi$ | $\Xi$ | $\Xi$ | $\Xi$ | $\Xi$ | $\Xi$ | $\Xi$ | $\Xi$ |

Table 8: The question level AUC and accuracy scores of AS2009, AL2005, BD2006 and NIPS34.

| | AUC | | | | Accuracy | | | |
|---|---|---|---|---|---|---|---|---|
| Model | Question Level(All-in-One) | | | | Question Level(All-in-One) | | | |
| | AS2009 | AL2005 | BD2006 | NIPS34 | AS2009 | AL2005 | BD2006 | NIPS34 |
| **DKT** | 0.7541±0.0011 | 0.8149±0.0011 | 0.8015±0.0008 | 0.7689±0.0002 | 0.7244±0.0014 | 0.8097±0.0005 | 0.8553±0.0002 | 0.7032±0.0004 |
| **DKT+** | 0.7547±0.0017 | 0.8156±0.0011 | 0.8020±0.0004 | 0.7696±0.0002 | 0.7248±0.0009 | 0.8097±0.0007 | 0.8553±0.0003 | 0.7039±0.0004 |
| **DKT-F** | - | 0.8147±0.0013 | 0.7985±0.0013 | 0.7733±0.0003 | - | 0.8090±0.0005 | 0.8536±0.0004 | 0.7076±0.0002 |
| **KQN** | 0.7477±0.0011 | 0.8027±0.0015 | 0.7936±0.0014 | 0.7684±0.0003 | 0.7228±0.0009 | 0.8025±0.0006 | 0.8532±0.0006 | 0.7028±0.0001 |
| **DKVMN** | 0.7473±0.0006 | 0.8054±0.0011 | 0.7983±0.0009 | 0.7673±0.0004 | 0.7199±0.0010 | 0.8027±0.0007 | 0.8545±0.0002 | 0.7016±0.0005 |
| **ATKT** | 0.7470±0.0008 | 0.7995±0.0023 | 0.7889±0.0008 | 0.7665±0.0001 | 0.7208±0.0009 | 0.7998±0.0019 | 0.8511±0.0004 | 0.7013±0.0002 |
| **GKT** | 0.7424±0.0021 | 0.8110±0.0009 | 0.8046±0.0008 | 0.7689±0.0024 | 0.7153±0.0032 | 0.8088±0.0008 | 0.8555±0.0002 | 0.7014±0.0028 |
| **SAKT** | 0.7246±0.0017 | 0.7880±0.0063 | 0.7740±0.0008 | 0.7517±0.0005 | 0.7063±0.0018 | 0.7954±0.0020 | 0.8461±0.0005 | 0.6879±0.0004 |
| **SAINT** | 0.6958±0.0023 | 0.7775±0.0017 | 0.7781±0.0013 | 0.7873±0.0007 | 0.6936±0.0034 | 0.7791±0.0016 | 0.8411±0.0065 | 0.7180±0.0006 |
| **AKT** | 0.7853±0.0017 | 0.8306±0.0019 | 0.8208±0.0007 | 0.8033±0.0003 | 0.7392±0.0021 | 0.8124±0.0011 | 0.8587±0.0005 | 0.7323±0.0005 |

Table 9: The KC level AUC and accuracy scores of AS2009, AL2005, BD2006 and NIPS34.

| | AUC | | | | Accuracy | | | |
|---|---|---|---|---|---|---|---|---|
| Model | KC Level(ALL-in-One) | | | | KC Level(ALL-in-One) | | | |
| | AS2009 | AL2005 | BD2006 | NIPS34 | AS2009 | AL2005 | BD2006 | NIPS34 |
| **DKT** | 0.7419±0.0011 | 0.8146±0.0016 | 0.8013±0.0008 | 0.7681±0.0002 | 0.7181±0.0014 | 0.7882±0.0011 | 0.8552±0.0002 | 0.7028±0.0004 |
| **DKT+** | 0.7424±0.0023 | 0.8144±0.0013 | 0.8019±0.0005 | 0.7689±0.0002 | 0.7191±0.0008 | 0.7889±0.0015 | 0.8552±0.0003 | 0.7034±0.0004 |
| **DKT-F** | - | 0.8163±0.0014 | 0.7984±0.0013 | 0.7727±0.0003 | - | 0.7891±0.0010 | 0.8535±0.0004 | 0.7071±0.0002 |
| **KQN** | 0.7361±0.0018 | 0.8005±0.0018 | 0.7935±0.0014 | 0.7677±0.0003 | 0.7179±0.0017 | 0.7850±0.0008 | 0.8532±0.0006 | 0.7023±0.0001 |
| **DKVMN** | 0.7330±0.0016 | 0.7891±0.0030 | 0.7981±0.0010 | 0.7668±0.0005 | 0.7144±0.0008 | 0.7778±0.0012 | 0.8544±0.0002 | 0.7013±0.0005 |
| **ATKT** | 0.7337±0.0014 | 0.7964±0.0043 | 0.7885±0.0008 | 0.7658±0.0001 | 0.7158±0.0012 | 0.7774±0.0030 | 0.8510±0.0004 | 0.7010±0.0002 |
| **GKT** | 0.7227±0.0013 | 0.8025±0.0034 | 0.8045±0.0008 | 0.7681±0.0025 | 0.7077±0.0018 | 0.7825±0.0023 | 0.8554±0.0002 | 0.7009±0.0029 |
| **SAKT** | 0.7085±0.0024 | 0.7682±0.0105 | 0.7738±0.0008 | 0.7516±0.0005 | 0.7017±0.0015 | 0.7729±0.0034 | 0.8460±0.0005 | 0.6878±0.0004 |
| **SAINT** | 0.6865±0.0024 | 0.6662±0.0099 | 0.7779±0.0013 | 0.7860±0.0011 | 0.6885±0.0037 | 0.7538±0.0011 | 0.8410±0.0065 | 0.7176±0.0006 |
| **AKT** | 0.7650±0.0012 | 0.8091±0.0030 | 0.8206±0.0008 | 0.8017±0.0006 | 0.7323±0.0026 | 0.7939±0.0016 | 0.8586±0.0004 | 0.7318±0.0006 |

Table 10: The overall AUC and accuracy scores of Statics2011, AS2015 and POJ.

| | AUC | | | Accuracy | | |
|---|---|---|---|---|---|---|
| Model | Statics2011 | AS2015 | POJ | Statics2011 | AS2015 | POJ |
| **DKT** | 0.8222±0.0013 | 0.7271±0.0005 | 0.6089±0.0009 | 0.7969±0.0006 | 0.7503±0.0003 | 0.6328±0.0020 |
| **DKT+** | 0.8279±0.0004 | 0.7285±0.0006 | 0.6173±0.0007 | 0.7977±0.0006 | 0.7510±0.0004 | 0.6482±0.0021 |
| **DKT-F** | 0.7839±0.0061 | - | 0.6030±0.0023 | 0.7872±0.0011 | - | 0.6371±0.0030 |
| **KQN** | 0.8232±0.0007 | 0.7254±0.0004 | 0.6080±0.0015 | 0.7978±0.0007 | 0.7500±0.0003 | 0.6435±0.0017 |
| **DKVMN** | 0.8093±0.0017 | 0.7227±0.0004 | 0.6056±0.0022 | 0.7929±0.0006 | 0.7508±0.0006 | 0.6393±0.0015 |
| **ATKT** | 0.8055±0.0020 | 0.7245±0.0007 | 0.6075±0.0012 | 0.7904±0.0011 | 0.7494±0.0002 | 0.6332±0.0023 |
| **GKT** | 0.8040±0.0065 | 0.7258±0.0012 | 0.6070±0.0036 | 0.7902±0.0021 | 0.7504±0.0010 | 0.6117±0.0147 |
| **SAKT** | 0.7965±0.0014 | 0.7114±0.0003 | 0.6095±0.0013 | 0.7879±0.0015 | 0.7474±0.0002 | 0.6407±0.0035 |
| **SAINT** | 0.7599±0.0139 | 0.7026±0.0011 | 0.5563±0.0012 | 0.7682±0.0056 | 0.7438±0.0010 | 0.6476±0.0003 |
| **AKT** | 0.8309±0.0009 | 0.7281±0.0004 | 0.6281±0.0013 | 0.8021±0.0011 | 0.7521±0.0005 | 0.6492±0.0010 |

information is available, the DLKT model performance is evaluated on both the original observed question-response data and the expended KC-response data. We conduct 5-fold cross validation for all the experiments and the averaged AUC/accuracy score and report the corresponding standard deviation.

## A.5   Problematic ATKT Original Implementation

There is an issue with the original ATKT implementation which utilizes the future ground truth to predict the mastery level of the current question. We compare the results of the original ATKT implementation (denoted as "ATKT-Wrong") and the fixed re-implementation, denoted as ATKT. Similar to Appendix A.4, Table 11 summarize the overall AUC performance results (averaged accuracy score from a 5-fold cross validation and the corresponding standard deviation scores) on both the original observed question-response data and the expended KC-response data in terms of AS2009, AL2005, BD2006, and NIPS34 datasets.

Table 11: The overall prediction performance in terms of AUC at both question level and KC level for ATKT-Wrong and ATKT.

| Model | Question Level(All-in-One) | | | | KC Level(ALL-in-One) | | | | Statics2011 | AS2015 | POJ |
|---|---|---|---|---|---|---|---|---|---|---|---|
| | AS2009 | AL2005 | BD2006 | NIPS34 | AS2009 | AL2005 | BD2006 | NIPS34 | | | |
| **ATKT-Wrong** | 0.7683 | 0.8082 | 0.7967 | 0.7804 | 0.7486 | 0.7944 | 0.7963 | 0.7792 | 0.8277 | 0.8172 | 0.6165 |
| **ATKT** | 0.7470 | 0.7995 | 0.7889 | 0.7665 | 0.7337 | 0.7964 | 0.7885 | 0.7658 | 0.8055 | 0.7245 | 0.6075 |

## A.6   Boosted DLKT Accuracy Results Due to Label Leakage

We provide the accuracy results of the one-by-one evaluation manner with label leakage problems, and report the accuracy exaggerated gains by computing the accuracy scores difference between results of one-by-one predictions at KC level and results of all-in-one predictions at KC level as shown in 12.

Table 12: The boosted DLKT accuracy results due to label leakage.

| Model | KC Level(One-by-One) | | | | Exaggerated Performance Gains ($\Delta$) | | | |
|---|---|---|---|---|---|---|---|---|
| | AS2009 | AL2005 | BD2006 | NIPS34 | AS2009 | AL2005 | BD2006 | NIPS34 |
| **DKT** | 0.7688 | 0.8701 | 0.8557 | 0.7069 | 0.0507 | 0.0819 | 0.0005 | 0.0041 |
| **DKT+** | 0.7694 | 0.8700 | 0.8557 | 0.7075 | 0.0503 | 0.0811 | 0.0005 | 0.0041 |
| **DKT-F** | - | 0.8701 | 0.8540 | 0.7112 | - | 0.0810 | 0.0005 | 0.0041 |
| **KQN** | 0.7659 | 0.8674 | 0.8537 | 0.7063 | 0.0480 | 0.0824 | 0.0005 | 0.0040 |
| **DKVMN** | 0.7650 | 0.8670 | 0.8548 | 0.7051 | 0.0506 | 0.0892 | 0.0004 | 0.0038 |
| **ATKT** | 0.7656 | 0.8648 | 0.8516 | 0.7050 | 0.0498 | 0.0874 | 0.0006 | 0.0040 |
| **GKT** | 0.7609 | 0.8696 | 0.8559 | 0.7050 | 0.0532 | 0.0871 | 0.0005 | 0.0041 |
| **SAKT** | 0.7410 | 0.8630 | 0.8462 | 0.6890 | 0.0393 | 0.0901 | 0.0002 | 0.0012 |
| **SAINT** | 0.7281 | 0.8563 | 0.8412 | 0.7205 | 0.0396 | 0.1025 | 0.0002 | 0.0029 |
| **AKT** | 0.7820 | 0.8768 | 0.8590 | 0.7360 | 0.0497 | 0.0829 | 0.0004 | 0.0042 |

## A.7   Detailed *AUC* Results (with Standard Deviations) of Performance Impacts on Different Lengths of Interaction History

In Section 4, we mention that different lengths of interaction history will influence the performance. We list the overall results of averaged AUC scores and the corresponding standard deviation scores on Table 13 and Table 14.

## A.8   Detailed *Accuracy* Results (with Standard Deviations) of Performance Impacts on Different Lengths of Interaction History

As mentioned in Section 4, the different lengths of interaction history will influence the performance. The overall results of averaged accuracy scores and the corresponding standard deviation scores are summarized in Table 15 and Table 16.

Table 13: AUC with standard deviation of different lengths in Statics2011, AS2009, AS2015 and AL2005.

| Model | Statics2011 | | AS2009 | | AS2015 | | AL2005 | |
|---|---|---|---|---|---|---|---|---|
| | L | S | L | S | L | S | L | S |
| **DKT** | 0.8219±0.0012 | 0.8314±0.0041 | 0.7351±0.0008 | 0.7650±0.0016 | 0.7106±0.0005 | 0.7281±0.0006 | 0.8160±0.0011 | 0.7623±0.0015 |
| **DKT+** | 0.8276±0.0005 | 0.8364±0.0034 | 0.7357±0.0020 | 0.7657±0.0018 | 0.7113±0.0005 | 0.7296±0.0006 | 0.8168±0.0011 | 0.7600±0.0021 |
| **DKT-F** | 0.7859±0.0057 | 0.7465±0.0134 | - | - | - | - | 0.8158±0.0013 | 0.7597±0.0011 |
| **KQN** | 0.8230±0.0006 | 0.8280±0.0038 | 0.7259±0.0016 | 0.7604±0.0007 | 0.7064±0.0011 | 0.7266±0.0004 | 0.8038±0.0015 | 0.7466±0.0034 |
| **DKVMN** | 0.8086±0.0017 | 0.8294±0.0043 | 0.7271±0.0018 | 0.7588±0.0006 | 0.7039±0.0009 | 0.7240±0.0004 | 0.8067±0.0012 | 0.7429±0.0016 |
| **ATKT** | 0.8046±0.0019 | 0.8295±0.0037 | 0.7249±0.0010 | 0.7605±0.0011 | 0.7029±0.0024 | 0.7262±0.0007 | 0.8004±0.0023 | 0.7564±0.0016 |
| **GKT** | 0.8044±0.0062 | 0.8004±0.0132 | 0.7224±0.0036 | 0.7535±0.0015 | 0.7111±0.0026 | 0.7266±0.0012 | 0.8122±0.0009 | 0.7528±0.0029 |
| **SAKT** | 0.7958±0.0015 | 0.8179±0.0048 | 0.6989±0.0026 | 0.7403±0.0014 | 0.6857±0.0005 | 0.7134±0.0005 | 0.7891±0.0064 | 0.7347±0.0045 |
| **SAINT** | 0.7592±0.0142 | 0.7845±0.0085 | 0.6687±0.0019 | 0.7112±0.0026 | 0.6617±0.0034 | 0.7060±0.0011 | 0.7788±0.0016 | 0.7097±0.0037 |
| **AKT** | 0.8305±0.0009 | 0.8466±0.0026 | 0.7781±0.0032 | 0.7878±0.0026 | 0.7113±0.0003 | 0.7292±0.0004 | 0.8317±0.0019 | 0.7771±0.0037 |

Table 14: AUC with standard deviation of different lengths in BD2006, NIPS34 and POJ.

| Model | BD2006 | | NIPS34 | | POJ | |
|---|---|---|---|---|---|---|
| | L | S | L | S | L | S |
| **DKT** | 0.8010±0.0008 | 0.8563±0.0065 | 0.7740±0.0002 | 0.7430±0.0005 | 0.5979±0.0011 | 0.6629±0.0015 |
| **DKT+** | 0.8015±0.0004 | 0.8593±0.0036 | 0.7748±0.0002 | 0.7436±0.0004 | 0.6045±0.0008 | 0.6782±0.0005 |
| **DKT-F** | 0.7980±0.0013 | 0.8467±0.0055 | 0.7784±0.0003 | 0.7480±0.0007 | 0.5915±0.0023 | 0.6606±0.0026 |
| **KQN** | 0.7931±0.0015 | 0.8515±0.0035 | 0.7738±0.0004 | 0.7414±0.0004 | 0.5944±0.0017 | 0.6774±0.0006 |
| **DKVMN** | 0.7978±0.0009 | 0.8540±0.0030 | 0.7725±0.0005 | 0.7414±0.0007 | 0.5924±0.0025 | 0.6732±0.0018 |
| **ATKT** | 0.7884±0.0008 | 0.8464±0.0054 | 0.7711±0.0002 | 0.7438±0.0004 | 0.5960±0.0015 | 0.6687±0.0013 |
| **GKT** | 0.8042±0.0008 | 0.8535±0.0040 | 0.7741±0.0024 | 0.7431±0.0028 | 0.5977±0.0043 | 0.6577±0.0039 |
| **SAKT** | 0.7734±0.0008 | 0.8239±0.0026 | 0.7570±0.0005 | 0.7253±0.0007 | 0.6001±0.0014 | 0.6544±0.0019 |
| **SAINT** | 0.7776±0.0014 | 0.8189±0.0056 | 0.7912±0.0007 | 0.7687±0.0011 | 0.5294±0.0015 | 0.6702±0.0005 |
| **AKT** | 0.8204±0.0007 | 0.8643±0.0026 | 0.8074±0.0003 | 0.7829±0.0005 | 0.6137±0.0016 | 0.6949±0.0006 |

Table 15: Accuracy with standard deviation of different lengths in Statics2011, AS2009, AS2015 and AL2005.

| Model | Statics2011 | | AS2009 | | AS2015 | | AL2005 | |
|---|---|---|---|---|---|---|---|---|
| | L | S | L | S | L | S | L | S |
| **DKT** | 0.7970±0.0007 | 0.7936±0.0022 | 0.7301±0.0014 | 0.7191±0.0015 | 0.7124±0.0010 | 0.7539±0.0003 | 0.8105±0.0006 | 0.7738±0.0053 |
| **DKT+** | 0.7976±0.0005 | 0.7996±0.0016 | 0.7316±0.0010 | 0.7186±0.0011 | 0.7136±0.0005 | 0.7546±0.0005 | 0.8106±0.0007 | 0.7720±0.0020 |
| **DKT-F** | 0.7877±0.0011 | 0.7745±0.0023 | - | - | - | - | 0.8098±0.0005 | 0.7759±0.0023 |
| **KQN** | 0.7979±0.0007 | 0.7980±0.0047 | 0.7303±0.0019 | 0.7160±0.0009 | 0.7118±0.0018 | 0.7536±0.0003 | 0.8032±0.0006 | 0.7717±0.0023 |
| **DKVMN** | 0.7924±0.0006 | 0.8041±0.0054 | 0.7279±0.0010 | 0.7125±0.0015 | 0.7144±0.0016 | 0.7542±0.0005 | 0.8035±0.0007 | 0.7706±0.0017 |
| **ATKT** | 0.7902±0.0013 | 0.7940±0.0030 | 0.7285±0.0006 | 0.7139±0.0015 | 0.7080±0.0019 | 0.7533±0.0002 | 0.8006±0.0019 | 0.7639±0.0020 |
| **GKT** | 0.7903±0.0018 | 0.7893±0.0083 | 0.7269±0.0028 | 0.7046±0.0058 | 0.7138±0.0002 | 0.7539±0.0010 | 0.8095±0.0008 | 0.7753±0.0024 |
| **SAKT** | 0.7873±0.0016 | 0.8021±0.0055 | 0.7126±0.0017 | 0.7004±0.0022 | 0.7052±0.0007 | 0.7514±0.0002 | 0.7961±0.0021 | 0.7640±0.0014 |
| **SAINT** | 0.7674±0.0056 | 0.7889±0.0072 | 0.7036±0.0036 | 0.6843±0.0034 | 0.7009±0.0021 | 0.7479±0.0013 | 0.7794±0.0016 | 0.7640±0.0030 |
| **AKT** | 0.8021±0.0010 | 0.8027±0.0032 | 0.7444±0.0017 | 0.7343±0.0031 | 0.7156±0.0017 | 0.7556±0.0006 | 0.8132±0.0011 | 0.7778±0.0039 |

Table 16: Accuracy with standard deviation of different lengths in BD2006, NIPS34 and POJ.

| Model | BD2006 | | NIPS34 | | POJ | |
|---|---|---|---|---|---|---|
| | L | S | L | S | L | S |
| **DKT** | 0.8554±0.0002 | 0.7961±0.0044 | 0.7078±0.0004 | 0.6820±0.0008 | 0.6246±0.0022 | 0.6810±0.0008 |
| **DKT+** | 0.8555±0.0003 | 0.7998±0.0044 | 0.7083±0.0004 | 0.6832±0.0006 | 0.6409±0.0024 | 0.6905±0.0011 |
| **DKT-F** | 0.8537±0.0004 | 0.7986±0.0036 | 0.7123±0.0002 | 0.6860±0.0005 | 0.6297±0.0036 | 0.6799±0.0014 |
| **KQN** | 0.8534±0.0006 | 0.7881±0.0030 | 0.7075±0.0002 | 0.6812±0.0006 | 0.6355±0.0020 | 0.6904±0.0009 |
| **DKVMN** | 0.8546±0.0002 | 0.7987±0.0038 | 0.7063±0.0005 | 0.6798±0.0005 | 0.6309±0.0017 | 0.6884±0.0015 |
| **ATKT** | 0.8513±0.0004 | 0.7769±0.0093 | 0.7051±0.0003 | 0.6838±0.0005 | 0.6241±0.0025 | 0.6867±0.0015 |
| **GKT** | 0.8556±0.0002 | 0.8004±0.0037 | 0.7058±0.0004 | 0.6813±0.0008 | 0.6001±0.0175 | 0.6794±0.0033 |
| **SAKT** | 0.8463±0.0005 | 0.7804±0.0024 | 0.6924±0.0004 | 0.6673±0.0008 | 0.6346±0.0041 | 0.6767±0.0016 |
| **SAINT** | 0.8414±0.0065 | 0.7598±0.0112 | 0.7217±0.0007 | 0.7011±0.0017 | 0.6399±0.0002 | 0.6926±0.0009 |
| **AKT** | 0.8588±0.0005 | 0.8121±0.0033 | 0.7366±0.0004 | 0.7127±0.0007 | 0.6401±0.0011 | 0.7023±0.0006 |

## A.9 Detailed Results of Performance Impacts on Different KC Prediction Fusion Mechanisms

The results of different KC prediction are shown on Table 17.

Table 17: Impact on different KC prediction fusion mechanisms

| Model | Dataset | Fusion Mechanisms | | | | Δ |
|---|---|---|---|---|---|---|
| | | LF-AVG | LF-MV | LF-S | EF | |
| DKT | AS2009 | 0.7541±0.0011 | 0.7526±0.0010* | 0.7524±0.0012* | - | 0.0015 |
| | AL2005 | 0.8149±0.0011 | 0.8123±0.0010* | 0.8131±0.0012* | - | 0.0018 |
| | BD2006 | 0.8015±0.0008 | 0.8015±0.0008○ | 0.8015±0.0008○ | - | 0.0000 |
| | NIPS34 | 0.7689±0.0002 | 0.7687±0.0002* | 0.7688±0.0002* | - | 0.0001 |
| DKT+ | AS2009 | 0.7547±0.0017 | 0.7533±0.0013* | 0.7530±0.0022* | - | 0.0014 |
| | AL2005 | 0.8156±0.0011 | 0.8124±0.0004* | 0.8146±0.0016* | - | 0.0010 |
| | BD2006 | 0.8020±0.0004 | 0.8020±0.0004○ | 0.8020±0.0004○ | - | 0.0000 |
| | NIPS34 | 0.7696±0.0002 | 0.7695±0.0002* | 0.7696±0.0002○ | - | 0.0000 |
| DKT-F | AS2009 | - | - | - | - | - |
| | AL2005 | 0.8147±0.0013 | 0.8122±0.0009* | 0.8131±0.0016* | - | 0.0016 |
| | BD2006 | 0.7985±0.0013 | 0.7985±0.0013○ | 0.7985±0.0013○ | - | 0.0000 |
| | NIPS34 | 0.7733±0.0003 | 0.7732±0.0003* | 0.7733±0.0003○ | - | 0.0000 |
| KQN | AS2009 | 0.7477±0.0011 | 0.7457±0.0013* | 0.7474±0.0012* | 0.7470±0.0011○ | 0.0003 |
| | AL2005 | 0.8027±0.0015 | 0.7985±0.0016* | 0.8012±0.0015* | 0.7935±0.0022○ | 0.0015 |
| | BD2006 | 0.7936±0.0014 | 0.7936±0.0014○ | 0.7936±0.0014○ | 0.7936±0.0014○ | 0.0000 |
| | NIPS34 | 0.7684±0.0003 | 0.7682±0.0003* | 0.7684±0.0003○ | 0.7684±0.0003○ | 0.0000 |
| DKVMN | AS2009 | 0.7473±0.0006 | 0.7458±0.0006* | 0.7456±0.0008* | 0.7454±0.0010* | 0.0015 |
| | AL2005 | 0.8054±0.0011 | 0.8022±0.0016* | 0.8021±0.0009* | 0.7961±0.0020* | 0.0032 |
| | BD2006 | 0.7983±0.0009 | 0.7983±0.0009○ | 0.7983±0.0009○ | 0.7983±0.0010○ | 0.0000 |
| | NIPS34 | 0.7673±0.0004 | 0.7672±0.0004* | 0.7673±0.0004○ | 0.7673±0.0004○ | 0.0000 |
| ATKT | AS2009 | 0.7470±0.0008 | 0.7440±0.0007* | 0.7466±0.0011* | - | 0.0004 |
| | AL2005 | 0.7995±0.0023 | 0.7963±0.0021* | 0.7974±0.0026* | - | 0.0021 |
| | BD2006 | 0.7889±0.0008 | 0.7888±0.0008* | 0.7889±0.0008○ | - | 0.0000 |
| | NIPS34 | 0.7665±0.0001 | 0.7663±0.0001* | 0.7665±0.0001○ | - | 0.0000 |
| GKT | AS2009 | 0.7424±0.0021 | 0.7376±0.0029* | 0.7401±0.002* | - | 0.0023 |
| | AL2005 | 0.8110±0.0009 | 0.8072±0.0008* | 0.8072±0.0012* | - | 0.0038 |
| | BD2006 | 0.8046±0.0008 | 0.8046±0.0008○ | 0.8046±0.0008○ | - | 0.0000 |
| | NIPS34 | 0.7689±0.0024 | 0.7686±0.0025* | 0.7689±0.0024○ | - | 0.0000 |
| SAKT | AS2009 | 0.7246±0.0017 | 0.7225±0.0020* | 0.7203±0.0016* | 0.7193±0.0021○ | 0.0021 |
| | AL2005 | 0.7880±0.0063 | 0.7801±0.0065* | 0.7859±0.0056* | 0.7697±0.0097* | 0.0021 |
| | BD2006 | 0.7740±0.0008 | 0.7739±0.0008* | 0.7740±0.0008○ | 0.7740±0.0008○ | 0.0000 |
| | NIPS34 | 0.7517±0.0005 | 0.7516±0.0005* | 0.7518±0.0005● | 0.7517±0.0005○ | -0.0001 |
| SAINT | AS2009 | 0.6958±0.0023 | 0.6957±0.0023* | 0.6957±0.0023* | 0.6957±0.0023○ | 0.0001 |
| | AL2005 | 0.7775±0.0017 | 0.7041±0.0133* | 0.7804±0.0037● | 0.6885±0.0145* | -0.0029 |
| | BD2006 | 0.7781±0.0013 | 0.7781±0.0013○ | 0.7781±0.0013○ | 0.7781±0.0013○ | 0.0000 |
| | NIPS34 | 0.7873±0.0007 | 0.7870±0.0008* | 0.7870±0.0008* | 0.7870±0.0009○ | 0.0002 |
| AKT | AS2009 | 0.7853±0.0017 | 0.7794±0.0009* | 0.7847±0.0021* | 0.7825±0.0026* | 0.0006 |
| | AL2005 | 0.8306±0.0019 | 0.8228±0.0022* | 0.8275±0.0019* | 0.8177±0.0026* | 0.0031 |
| | BD2006 | 0.8208±0.0007 | 0.8208±0.0007○ | 0.8208±0.0007○ | 0.8208±0.0007○ | 0.0000 |
| | NIPS34 | 0.8033±0.0003 | 0.8028±0.0005* | 0.8033±0.0003○ | 0.8034±0.0003● | -0.0001 |
| #win/#tie/#loss | | - | 31/8/0 | 20/17/2 | 6/13/1 | |

## A.10 Detailed *AUC* Results of Performance Impacts on Accumulative and Non-Accumulative Prediction Settings

In Section 4, we show the AUC performance of accumulative and non-accumulative prediction settings on 5 classic models with various category DLKT methods (i.e., DKT, DKVMN, ATKT, GKT and AKT) on AS2009, BD2006 and POJ datasets. The AUC scores for 5 classic models and 10 models of our benchmark on all 7 datasets are shown in Figure 5 and 6 respectively.

## A.11 Detailed *Accuracy* Results of Performance Impacts on Accumulative and Non-Accumulative Prediction Settings

Similar to Appendix A.10, the accuracy scores for 5 classic models and 10 models of our benchmark on all 7 datasets are shown in Figure 7 and 8 respectively.

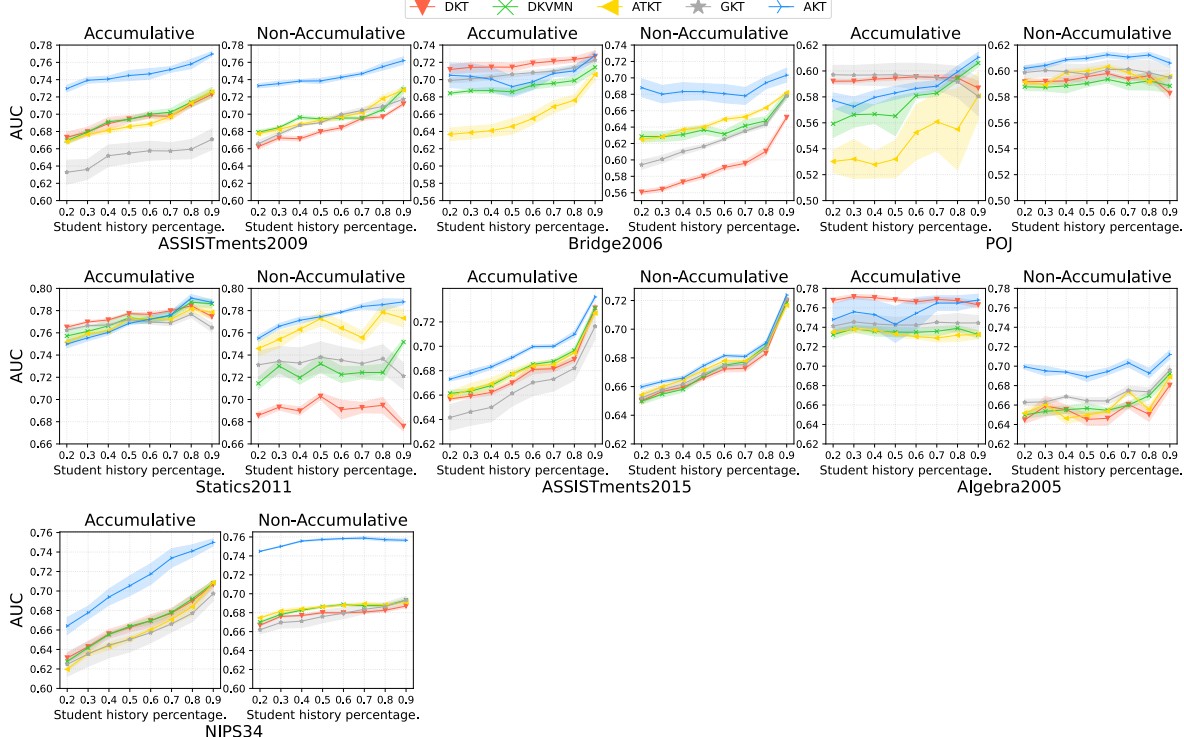

Figure 5: AUC of multi-step prediction for all datasets (5 models).

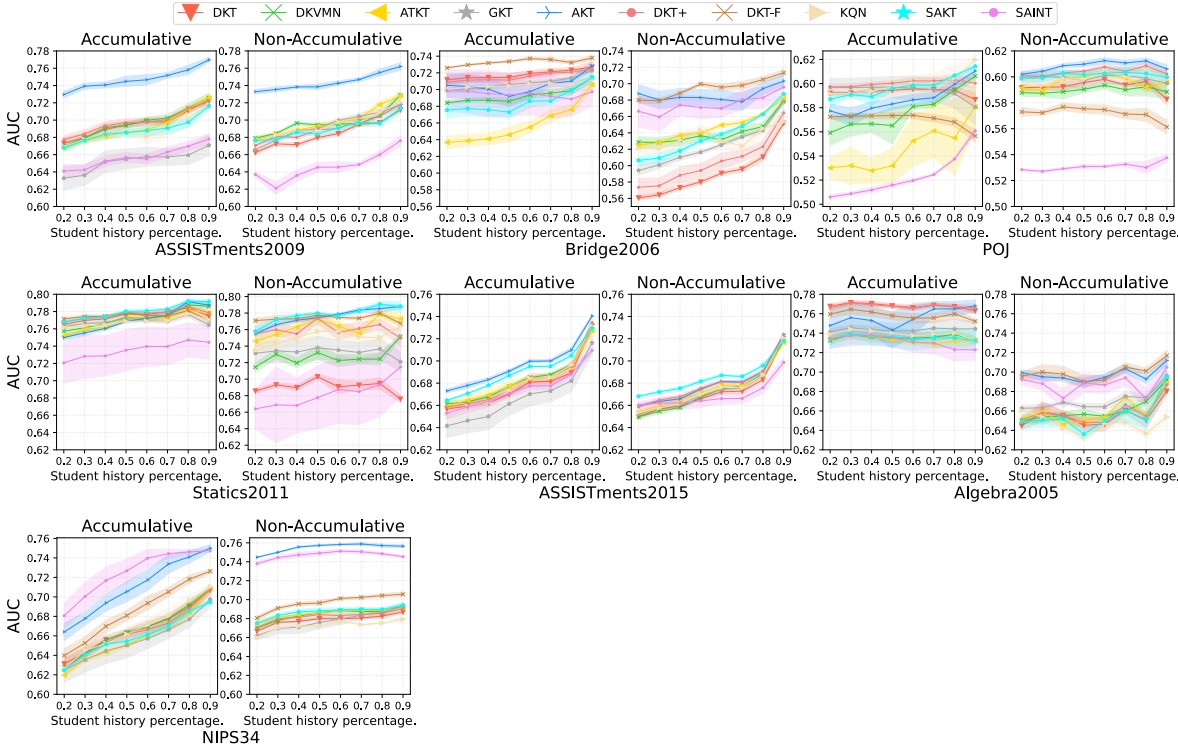

Figure 6: AUC of multi-step prediction for all datasets (10 models).

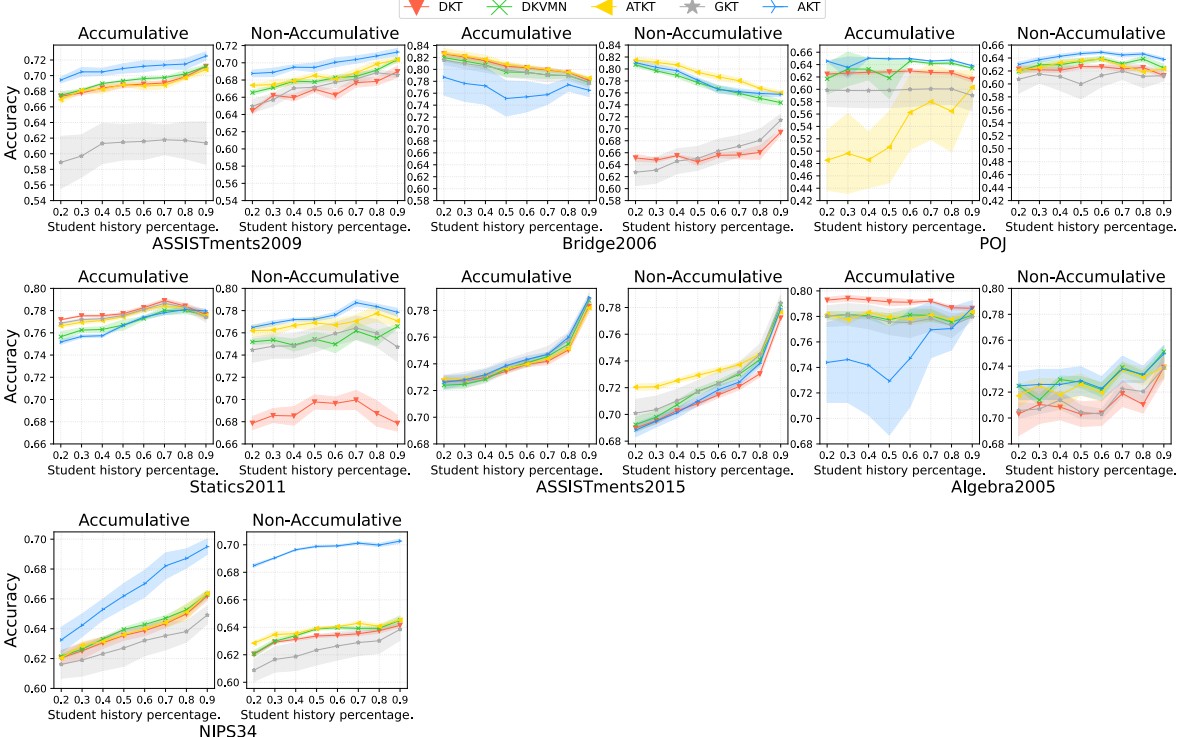

Figure 7: Accuracy of multi-step prediction for all datasets (5 models).

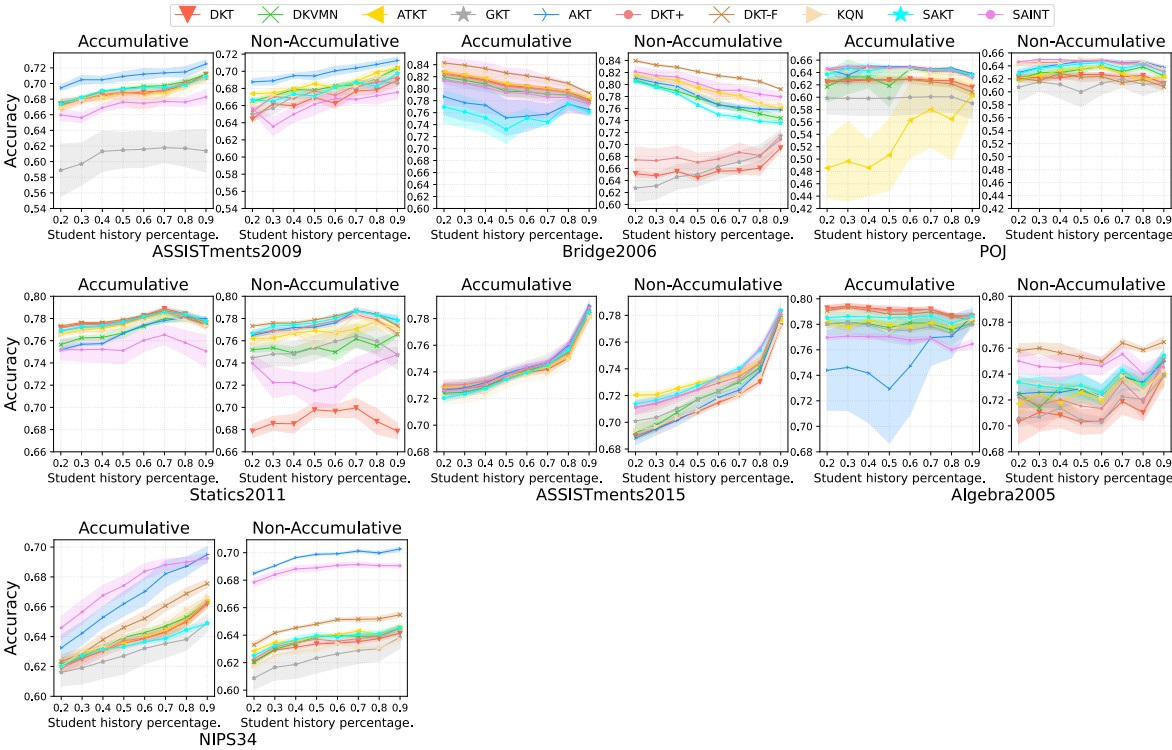

Figure 8: Accuracy of multi-step prediction for all datasets (10 models).