# OpenReview forum: "pyKT: A Python Library to Benchmark Deep Learning based Knowledge Tracing Models"
_NeurIPS.cc/2022/Track/Datasets_and_Benchmarks — NeurIPS 2022 Datasets and Benchmarks _

### Official Review · Reviewer_q5NU · 2022-07-16
**Comprehensive comparison across a representative sample of DLKT methods and datasets**

**Rating:** 8
**Confidence:** 3

**Strengths:**

- a comprehensive comparison across a representative sample of deep learning-based knowledge tracing (DLKT) models
- this comparison yields several worrying findings, such as the existence of data leakage artificially increasing performance metrics
- publically release a benchmark to standardize evaluation for future work in DLKT methods

**Weaknesses:**

At least for English, the GitHub repository does not appear to have any instructions beyond installation (see "Documentation" below). Docking a point for this, but would be happy to raise my score if the authors agree to add these instructions.

**Additional Feedback:**

- It looks like the PDF was rendered without line numbers? This makes our job as reviewers harder :(

**Clarity:**

In general, I found that the paper was well written. Here are a few small mistakes/typos and points of clarity I noticed:

- Abstract ~line8: "and/or" is frowned upon in formal writing. I would choose one or the other.
- In the second paragraph of the introduction, "knowledge components" are mentioned but not defined. Might be good to define this for readers less familiar with the knowledge tracing literature.
- In the third paragraph of the introduction, its mentioned that: "evaluations of existing DLKT work are not standardized and experimental results of the same approach on the same dataset vary surprisingly from 0.709 to 0.86". I don't think there's enough context here. What is the metric being quoted? What is causing this range, e.g. different random initializations? Is this the delta between the methods reported score and someone else's reproduction? I eventually found some more context in section 3.3, so at the very least, I would cite this section here.

**Correctness:**

The evaluation and benchmark construction appear sound to me. However, I admit that I am less familiar with the area of DLKT, so perhaps other reviewers will be able to comment with more depth on correctness.

**Documentation:**

The dataset creation process is described, and the authors link to a public GitHub repository. However,

- I don't see a maintenance plan
- At the time of writing this review, I don't see any documentation for the benchmark on GitHub (https://github.com/pykt-team/pykt-toolkit) or the website (https://pykt.org/) beyond installation instructions in English. Would want to see some basic usage instructions at a minimum. Some worked examples would be even better.

**Ethics:**

There is no discussion in the paper about ethical concerns. Although I don't think there are any glaring ethical concerns, because this work ultimately pertains to applying predictive models to education, it would have been nice to see a short discussion as to the harms that mistakes by a deployed model could cause.

**Relation To Prior Work:**

The paper heavily sites previous methods and datasets in the area of KT and DLKT. The authors also justify not including some DLKT methods that use auxiliary information (while still citing them). Although I am not very familiar with the KT literature, it appears that the authors were careful to cite the majority of existing, relevant literature and place their contributions in context.

**Summary And Contributions:**

This paper conducts a comprehensive comparison across a representative sample of deep learning-based knowledge tracing (DLKT) models, producing several insightful (if worrying!) findings, including that recently proposed methods hardly outperform one of the first DLKT methods published seven years ago. I think this on its own is a major contribution to this area of research, and so I am recommending acceptance.

---

### Official Review · Reviewer_hDCb · 2022-07-20
**A useful library to compare common DLKT methods**

**Rating:** 7
**Confidence:** 3

**Strengths:**

- The authors tackle an often neglected but still extremely important aspect which is reproducibility of existing methods.
- They provide opensource and *on-the-shelf* implementations of existing methods together with ready-to-use data pre-processing, data splitting and model evalutation protocols. This work should ease reproducibility and reduce barriers for new methods to compare to existing ones.
- The benchmark are well conducted with extensive hyper-parameter search.

**Weaknesses:**

- The code is documented with an online documentation but I think some work remains to be done on this documentation to easily use this software. For instance, there is no *quickstart* available in English and many of the methods are not documented. I am afraid this is a brake to make the code easily usable. Maybe the authors can consider adding some notebooks to illustrate a basic usage of their software as well.
- The authors states that they are open to new contributors and so I suggest they polish their documentation and include some guidelines for new comers to know how to easily integrate their models and contribute.
- The repository does not include any unit-testing tool allowing continuous integration.

**Additional Feedback:**

- Could you kindly provide the **Checklist** that is a required appendix?
- Can you provide a clear definition of what should be understood by *knowledge component*?

**Clarity:**

The paper is well written and easy to follow. I nonetheless think  the authors should sometime use a slightly softer tone when discussing peers methods (e.g. "mysterious",  "ironically", "such steps are *vaguely* described in existing publications"). This would strengthen the point they are making.

**Correctness:**

The benchmark is well conducted with an extensive hyperparameter search and using 5-fold cross validation. The range of parameters considered is provided and favor reproducibility.

**Documentation:**

An online documentation is available, but this should be polished a bit to make the software easier to use.

**Ethics:**

The authors provide the link to the raw datasets they use but maybe the associated licence should be specified.

**Relation To Prior Work:**

To best of my knowledge, the authors position well their paper among existing works and cite any relevant publications.

**Summary And Contributions:**

This paper introduces pyKT, a python library providing implementations of 10 Deep Learning based Knowledge Tracing (DLKT) models. This library also includes standardized data pre-processing methods and evaluation protocols allowing to benchmark the main deep learning based approaches to perform Knowledge Tracing. The paper also proposes to perform a benchmark of these models on 7 datasets.

---

### Official Review · Reviewer_VXMQ · 2022-07-26
**pyKT library to benchmark DLKT models**

**Rating:** 7
**Confidence:** 3
**Correctness:** The benchmark design sounds reasonable.
**Clarity:** The paper is very well written and cl…

**Strengths:**

The problem and the contribution are important and explained very clearly.
The authors propose a platform which helps different methods to be compared and evaluated in a more standardized way.
They worked on 7 popular datasets in KT and 10 state-of-the-art DLKT models.
Their open-source library, which includes some DLKT implementations and evaluation protocols, can be valuable for future research.



**Weaknesses:**

Some minor comments:
The authors discussed about evaluation protocols, comparison of different models, and so on but I do not see a very concrete discussion regarding scalability, which could be nice to include.
Table 1 basically repeats some of the statistics in the text which could be avoided.

**Additional Feedback:**

N/A

**Documentation:**

The link to their open source library is available. Also, sufficient details is available in their paper and provided documents.

**Ethics:**

I do not have any ethical concerns.

**Relation To Prior Work:**

Yes, it is clearly discussed and the authors comprehensively studied previous work.

**Summary And Contributions:**

The authors state that data preprocessing procedures in existing DLKT approaches are often private or custom and they differ in terms of the evaluation protocol. To address these and make valid comparisons across DLKT methods happen, they introduced pyKT library. It consists standardized preprocessing procedures on 7 popular datasets across different domains and 10 state-of-the-art model implementations. The authors provided 5 observations and suggestions from their results. One of these observations suggests that wrong evaluation setting may cause label leakage that generally leads to performance inflation.

---

### Official Review · Reviewer_BCcA · 2022-07-27
**Review for PyKT**

**Rating:** 7
**Confidence:** 3
**Correctness:** The correctness seems good.
**Clarity:** Seems good.

**Strengths:**

1. This paper presents a problem statement section, which is reader-friendly.
2. This work is overall well-motivated, and discuss its application in real-world scenarios.
3. The authors present detailed analyses and insights into the experimental results.

**Weaknesses:**

1. Potential limitions are not discussed in this paper.
2. The authors should provide more instructions on how to contribute to the PyKT.

**Additional Feedback:**

I don't have more comments.

**Documentation:**

The presented PyKT toolkit seems well-documented and maintenance well.

**Ethics:**

I don't think there are any ethics issues in this work.

**Relation To Prior Work:**

Good, the authors discuss the difference from prior works and address their contributions.

**Summary And Contributions:**

This work focuses on the knowledge tracing problem. The authors point out that the preprocess procedures in existing works are often private and the evaluation protocols are different and far away from the real world senerio. To address this issues, this paper presents a comprehensive python toolkit named pyKT. The toolkit provides standardized dataset preprocess procedure and several popular DLKT model implementations.

---

### Official Review · Reviewer_tP7n · 2022-07-27
**A useful python library and well-written paper.**

**Rating:** 7
**Confidence:** 3
**Clarity:** This paper is well-written.

**Strengths:**

1. The proposed benchmark plated-form PYKT is novel and can guarantee valid comparisons across DLKT methods via thorough evaluations.

2. Experimental results on the fine-grained and rigorous empirical KT studies yield a set of observations and suggestions for effective DLKT.

3. This paper provides comprehensive experiments with insightful analysis.

4. I read some codes in the toolkit and found it is well-organized.

**Weaknesses:**

1. I  think more background about knowledge tracing baselines will be helpful.

**Additional Feedback:**

No.

**Correctness:**

The claims in this paper are correct, and the datasets are constructed in a sound way. The evaluation methods and experiment design are correct.

**Documentation:**

Yes.

**Ethics:**

This paper is well-written and easy to follow.

**Relation To Prior Work:**

Yes.

**Summary And Contributions:**

This paper proposes a comprehensive python-based benchmark platform, PYKT, to guarantee valid comparisons across DLKT methods via thorough evaluations. The PYKT library consists of a standardized set of integrated data preprocessing procedures on 7 popular datasets across different domains and 10 frequently compared DLKT model implementations for transparent experiments. Experimental results on the fine-grained and rigorous empirical KT studies yield a set of observations and suggestions for effective DLKT. The proposed toolkit is open source. Overall, I believe this paper has made a good contribution to the knowledge tracing community, although I am not an expert in this area, I find this paper easy to follow.

---

### Official Review · Reviewer_ZCRa · 2022-07-28
**Very thoughtful and thorough approach for creating a benchmark for deep learning knowledge tracing models**

**Rating:** 9
**Confidence:** 4

**Strengths:**

Whatever one may say about the particulars of this benchmark, the team is tackling a very important challenge within the research community: how can we ensure that we are comparing apples-to-apples and that the results being reported are reproducible and meaningful. The literature is filled with papers where someone tweaks a current model in some small way and reports a marginal gain in performance that is elusive to replicate.

The strength of this paper is the thoroughness of this first step towards creating a community benchmark. The team did a terrific job reviewing the plethora of models out there and picking core, representative examples along with selecting appropriate and available data sets. The paper is framed around two well-chosen research questions that motivate the concluding observations.


**Weaknesses:**

The limitation of this work applies to all knowledge tracing approaches that reduce student learning assessment down to a binary value of correct and incorrect. We know that these sorts of observations usually measure recall or recognition, rather than deep comprehension by the students themselves. Contemporary approaches towards assessment will need to handle longer and more nuanced language and require much better NLP. But, all of the datasets that are being used are based on this binary response assumption.

The paper makes a big deal about label leakage but it could have been explained much more clearly!


**Additional Feedback:**

None

**Clarity:**

The paper is well written except for the caveat that a better explanation of the label leakage issue is warranted. But, this appears to be mainly driven by the page limits.

**Correctness:**

The benchmark experiments are well thought out with appropriate measures being used.  It should be straightforward for another group to use this toolkit.

**Documentation:**

There is sufficient detail for other groups to use this benchmark and attendant toolkit.

**Ethics:**

They are using available datasets from ASSISTments, NeurIPS and EDM for the most part. It is unknown what consent procedures were used for the Peking coding practice online platform.  But, this dataset ws used in prior peer-reviewed publications.

**Relation To Prior Work:**

The grounding in prior work is a real strength. The authors reviewed a lot of existing DLKT models and classified them into major categories, selecting representative models from each category to reimplement/test. The datasets selected for the experiments were appropriate justified and adequately described.

**Summary And Contributions:**

This paper describes a new benchmark for comparing and contrasting deep learning knowledge tracing models. This benchmark is motivated by the observation that the research community is busy creating a variety of models based on different assumptions and approaches, some major and some minor, and that the results on standard data sets vary in ways that make judging new models difficult. The paper posits that data cleansing and other pipeline processes may explain the variability rather than factors intrinsic to the model being proposed. The chief contribution is a toolkit called pyKT, which include python routines for standardizing data cleansing and data set preparation, as well as attendant recommended procedures. The benchmark was utilized to compare 5 different flavors of DLKT models across 7 different publicly available datasets. The evaluation tasks included predicting the student’s response on the last question based on historical data and predicting multiple student responses. In all cases, the student response is a binary variable of either correct or incorrect. The paper concludes with a series of observations on the performance of the selected models on these data sets using the proposed benchmark toolkit: the chief finding is that the original DKT model still performs the best.

---

### Meta-Review · Area_Chair_yPR6 · 2022-09-11

**Recommendation:** Accept
**Confidence:** 4

**Metareview:**

This work develops a python-based benchmark platform (PYKT) that implements several Deep Learning based Knowledge Tracing models. The research is well motivated and the paper is well written. The experimental section is also thorough and provides procedures for handling several popular datasets across different domains. The reviewers raised some minor concerns, and the authors are requested to address them in their final submission.

---

### Decision · Program_Chairs · 2022-09-16

Accept